# Combined transient ablation and single-cell RNA-sequencing reveals the development of medullary thymic epithelial cells

Kristen L Wells[1†], Corey N Miller[2,3†], Andreas R Gschwind[1], Wu Wei[4], Jonah D Phipps[2,3], Mark S Anderson[2,3*], Lars M Steinmetz[1,4,5*]

[1]Department of Genetics, Stanford University School of Medicine, Stanford, United States; [2]Diabetes Center, University of California, San Francisco, San Francisco, United States; [3]Department of Medicine, University of California San Francisco, San Francisco, United States; [4]Stanford Genome Technology Center, Stanford University, Palo Alto, United States; [5]Genome Biology Unit, European Molecular Biology Laboratory (EMBL), Heidelberg, Germany

**Abstract** Medullary thymic epithelial cells (mTECs) play a critical role in central immune tolerance by mediating negative selection of autoreactive T cells through the collective expression of the peripheral self-antigen compartment, including tissue-specific antigens (TSAs). Recent work has shown that gene-expression patterns within the mTEC compartment are heterogenous and include multiple differentiated cell states. To further define mTEC development and medullary epithelial lineage relationships, we combined lineage tracing and recovery from transient in vivo mTEC ablation with single-cell RNA-sequencing in *Mus musculus*. The combination of bioinformatic and experimental approaches revealed a non-stem transit-amplifying population of cycling mTECs that preceded *Aire* expression. We propose a branching model of mTEC development wherein a heterogeneous pool of transit-amplifying cells gives rise to *Aire-* and *Ccl21a*-expressing mTEC subsets. We further use experimental techniques to show that within the *Aire*-expressing developmental branch, TSA expression peaked as *Aire* expression decreased, implying *Aire* expression must be established before TSA expression can occur. Collectively, these data provide a roadmap of mTEC development and demonstrate the power of combinatorial approaches leveraging both in vivo models and high-dimensional datasets.

**\*For correspondence:**
Mark.Anderson@ucsf.edu (MSA);
lars.steinmetz@embl.de (LMS)

[†]These authors contributed equally to this work

**Competing interests:** The authors declare that no competing interests exist.

## Introduction

The thymus is the primary site of T-cell development and is required for the establishment of central immune tolerance (*Klein et al., 2014*). Within the thymus, medullary thymic epithelial cells (mTECs) play a key role in enforcing tolerance by expressing a wide array of tissue-specific self-antigens (TSAs) that serve to eliminate potentially autoreactive developing T cells (*Derbinski et al., 2008*; *Lancaster et al., 2019*; *Metzger and Anderson, 2011*). The autoimmune regulator (Aire) protein plays a central and nonredundant role in the expression of a subset of TSAs (*Anderson et al., 2002*) and is believed to function, in part, by modulating chromatin accessibility (*Koh et al., 2018*; *Perniola, 2018*). The transcription factor Fezf2 has also been shown to promote the expression of TSAs within mTECs in an Aire-independent manner (*Takaba et al., 2015*). *Aire*-expressing mTECs are characterized by high levels of MHC class II expression in the adult thymus and have an estimated half-life of 12–14 days (*Gray et al., 2006*). These mTEC-MHC Class II high cells (mTEC-hi) are thought to arise from mTEC-MHC Class II low progenitors (mTEC-lo) (*Gray et al., 2007*) through

**eLife digest** Specialized cells in the immune system known as T cells protect the body from infection by destroying disease-causing microbes, such as bacteria or viruses. T cells use proteins on their surface called receptors to stick to infectious microbes and remove them from the body. Some newly developed T-cells, however, contain receptors that recognize and bind to cells that belong in the body. If these faulty T cells are released, they can attack healthy tissues and cause an autoimmune disease.

After a new T cell is developed, it gets carried to a gland in the chest known as the thymus. Cells in the thymus called mTECs screen T cells for receptors that may bind to the body's tissues. mTECs do this by presenting T cells with proteins that are commonly found on the surface of healthy cells in the body. If a T cell recognizes any of these 'tissue specific proteins', it is destroyed or given a new role in the body. Some faulty T cells, however, still manage to evade detection. One way to uncover why this might happen is to investigate how mTECs develop. Previous work showed that mTECs transition through various stages before reaching their final form. However, the order in which these events occur remained unclear.

To gain a better understanding of these developmental steps, Wells, Miller et al. extracted mTECs from the thymus of mice and analyzed the genetic make-up of individual cells. This uncovered a missing link in mTEC development: a new type of cell that is the immediate predecessor of the final mTEC. These 'predecessor' cells were actively growing, highlighting that mTECs can be constantly generated in the body. By probing the genes that generate tissue-specific proteins in mTECs, Wells, Miller et al. revealed that these proteins were only produced for short periods and in the late stages of mTEC development.

These findings contribute to our understanding of how mTECs develop to screen T cells. Mapping these developmental stages will make it easier to identify when faulty T cells are able to evade mTECs. This will lead to earlier detection of autoimmune diseases which could result in better treatments.

inductive signals that include signaling through receptor activator of NF-κB (RANK) (*Rossi et al., 2007*). Within mTEC-hi cells, single-cell studies have shown that expression of TSAs is highly heterogeneous, with each cell expressing only 1–3% of all TSAs (*Anderson and Jenkinson, 2015*; *Brennecke et al., 2015*; *Meredith et al., 2015*). This heterogeneity is likely due to complex regulatory events that occur before expression of *Aire* and *Fezf2*.

Considerable effort has been devoted to identifying the progenitors of *Aire*-expressing mTECs in the mTEC-lo compartment. It has been suggested that these progenitors express high levels of *Ccl21a* (*Onder et al., 2015*), which encodes a chemokine ligand known to regulate the migration of positively selected thymocytes into the thymic medulla (*Kozai et al., 2017*). However, disruption of these cells in mice without lymphotoxin beta receptor (LTBR) does not impact *Aire*-expressing mTECs, implying that *Ccl21a*-expressing cells may not be the precursors of *Aire*-expressing mTECs (*Lkhagvasuren et al., 2013*). Additionally, thymic cells expressing Sca-1 were recently found to self-renew and generate both mTEC and cortical thymic epithelial cell (cTEC) lineages, suggesting these may be a bi-potent progenitor of early TEC development and further obfuscating previous work describing epithelial subset relationships (*Ucar et al., 2014*; *Wong et al., 2014*).

Recently, single-cell RNA-sequencing (scRNA-seq) has been utilized to define the mTEC compartment during neonatal and adult stages of development (*Dhalla et al., 2020*; *Kernfeld et al., 2018*; *Miragaia et al., 2018*). Using this approach, we and others identified a previously unappreciated and highly-differentiated mTEC subset that bore striking similarity to peripheral tuft cells found at mucosal barriers (*Bornstein et al., 2018*; *Miller et al., 2018*). Although scRNA-seq is a powerful tool for characterizing distinct cell populations, including minor and rare populations (*Hwang et al., 2018*), assigning precursor–product relationships is difficult based on gene expression alone. In the current manuscript, we reasoned that combining scRNA-seq with in vivo tools would better resolve our understanding of the developmental relationships between mTECs. First, we used a bioinformatic approach on scRNA-seq of control thymus to characterize mTEC lineage relationships. Second, we combined scRNA-seq with a lineage tracing approach (*Kretzschmar and Watt, 2012*) in

which the developmental relationships of *Aire*-expressing mTECs and their progeny are readily discernible (*Metzger et al., 2013*). Third, we combined scRNA-Seq with a transient ablation model in which differentiation of mTEC-hi cells is selectively blocked after treatment with anti-RANK-ligand (RANKL) antibody. Inhibition of RANK signaling depletes the *Aire*-expressing mTEC population, which then recovers over 10 weeks (*Khan et al., 2014*; *Metzger et al., 2013*). Using these approaches, we created a molecular roadmap of *Aire*-expressing mTEC development that details the kinetics of TSA expression in relation to Aire and identified a transit-amplifying population of mTECs that we propose is the immediate precursor of the *Aire*-expressing and *Ccl21a*-expressing populations.

## Results

### Single-cell classification of mTEC populations

First, we utilized a bioinformatic approach to identify TEC populations and lineage relationships at the single-cell level that could be subsequently validated using lineage tracing and transient in vivo ablation (*Figure 1a*). We performed single-cell transcriptome analysis of total TECs purified by flow cytometry (EpCAM$^+$ CD45$^-$) and subjected to scRNA-seq on the 10x Genomics platform (*Zheng et al., 2017*; *Figure 1a* and *Figure 1—figure supplement 1a*). A total of 2434 cells were included in the downstream analysis.

Single cells were projected into a reduced-dimensional space using UMAP and were clustered based on the top 13 principle components, yielding six total clusters (*Adey, 2019*; *Becht et al., 2018*; *Butler et al., 2018*; *Figure 1b* and *Figure 1—figure supplement 1b*). To identify clusters, we performed differential gene-expression analysis and searched for gene signatures previously associated with thymic epithelial subsets. As described in previous TEC single-cell studies, the 'cTEC' cluster was marked by high expression of *Ackr4* and *Prss16* (*Bornstein et al., 2018*; *Kernfeld et al., 2018*; *Miragaia et al., 2018*; *Figure 1c and d*, *Figure 1—figure supplement 1c*). The '*Ccl21a*-high' cluster was marked by high expression of *Ccl21a* (*Lkhagvasuren et al., 2013*) and *Krt5* (*Figure 1c and d*, *Figure 1—figure supplement 1c*). The gene-expression pattern within the '*Ccl21a*-high' cluster was consistent with populations that have previously been described as jTEC (*Miragaia et al., 2018*), mTEC2 (*Kernfeld et al., 2018*), and mTEC-I (*Bornstein et al., 2018*). The 'Tuft' cluster had high expression of *Trpm5*, *Dclk1*, and a gene-expression pattern that closely agreed with previous single-cell descriptions of thymic tuft cells (*Bornstein et al., 2018*; *Miller et al., 2018*; *Figure 1c and d*, *Figure 1—figure supplement 1c*). The remaining three clusters, dubbed 'TAC-TEC', '*Aire*-positive', and 'Late-*Aire*' based on subsequent analysis, included cells that expressed *Aire* and *Fezf2* and resembled populations previously described as mTEC3 (*Kernfeld et al., 2018*), mTEC-II and mTEC-III (*Bornstein et al., 2018*), and mTEhi and mTEClo (*Miragaia et al., 2018*; *Figure 1c and d*, *Figure 1—figure supplement 1c*). As further validation of our clustering, we restricted the list of differentially expressed genes to only those annotated as transcription factors and chromatin modifiers because these functional categories are known to be important in TEC biology (*Wang et al., 2019*; *Figure 1—figure supplement 1d*). We anticipated that differential expression analysis between clusters would be enriched for specific genes previously associated with mTEC function. Reassuringly, *Aire* (*Ramsey et al., 2002*), *Ptma* (*Moretti et al., 2015*), *Hmgb1* (*Guha et al., 2017*), *Irf7* (*Otero et al., 2013*), *Cited2* (*Michell et al., 2010*), *Pax1* (*Romano et al., 2013*), and *Spib* (*Akiyama et al., 2014*) were all found in the differentially expressed gene set and have known knockout phenotypes in mice affecting thymus development and function (*Figure 1—figure supplement 1d*).

### Single-cell velocity analysis suggests that the 'TAC-TEC' cluster is a precursor of most mTEC subpopulations

Next, we used computational predictive algorithms to define the lineage relationships between the identified mTEC subpopulations. To accomplish this, we applied single-cell velocity (*Bergen et al., 2019*), the stochastic version of transcriptional dynamics used in RNA velocity, which uses intron retention to predict the future transcriptional state of a single cell and infer lineage relationships between cell populations (*La Manno et al., 2018*). Application of single-cell velocity to our data predicted that 'TAC-TECs' preceded the '*Aire*-positive' population and that the '*Aire*-positive'

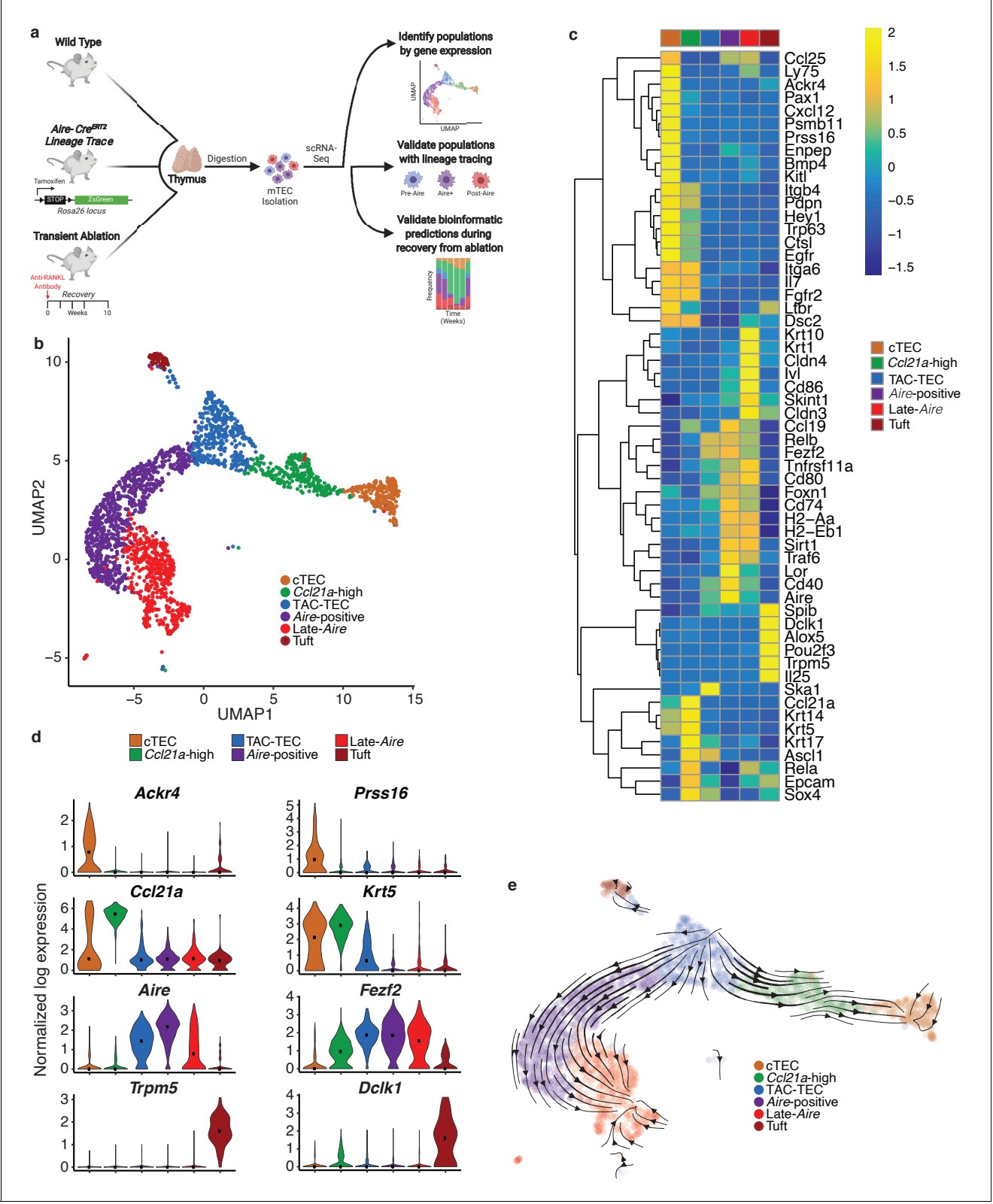

**Figure 1.** Single-cell sequencing of medullary thymic epithelial cells reveals population relationships. (A) Outline of experimental design. Lineage relationships were first determined bionformatically from single-cell RNA-sequencing of control thymus. Lineage relationships and gene-expression events were then validated using both an inducible in vivo lineage tracing system and a model of transient thymus ablation and recovery. (B) UMAP visualization of all 2434 cells and clustering (color) of mTECs (two replicates, three pooled thymi each). Each point represents a cell. (C) Heatmap of

*Figure 1 continued on next page*

*Figure 1 continued*

genes curated from the literature across mTEC populations identified in this study. Color is mean-centered, log-normalized expression. Gene expression is the average expression within each mTEC population. (D) Normalized log expression of marker genes for known mTEC populations across cells in each cluster. Color represents cluster, black dot is median expression. (E) Single-cell velocity plot showing lineage relationships between mTECs projected onto a UMAP dimensional reduction. Arrows represent predicted developmental trajectories. See also *Figure 1—figure supplement 1*.

The online version of this article includes the following figure supplement(s) for figure 1:

**Figure supplement 1.** Classification of medullary thymic epithelial cell populations.

population preceded the 'Late-*Aire*' population (*Figure 1e*). Further, single-cell velocity predicted a branching model of mTEC development in which 'TAC-TECs' give rise to both the '*Aire*-positive' and '*Ccl21a*-high' populations (*Figure 1e*), rather than a precursor-product relationship between *Ccl21a*-expressing and *Aire*-expressing mTECs (*Onder et al., 2015*). Additionally, this model also predicted a path to minor mTEC subsets, including the 'Tuft' cluster. Therefore, single-cell velocity analysis suggested that the 'TAC-TEC' cluster populates the mTEC subsets observed in this study.

## The TAC-TEC population is actively cycling

To further characterize the 'TAC-TEC' cluster, we generated an unbiased list of all genes specifically upregulated in these cells (*Figure 2—figure supplement 1a*). We noticed that this restricted list included a preponderance of chromatin-modifying factors, including *Hmgb2*, *H2afz*, *Hmgn2*, *Hmgb1*, and *Hmgn1* (*Figure 2—figure supplement 1a*). Because many of these genes were also upregulated in other mTEC populations, we restricted this analysis to include only those genes uniquely upregulated in 'TAC-TECs' (35 total genes). It had been hypothesized that a transit-amplifying cell exists in the thymus (*Gray et al., 2007*) therefore, we compared the transcriptional signature of the 'TAC-TECs' to the gene signature of a published transit-amplifying population (*Basak et al., 2018*) and noticed a striking overlap. To test this overlap, we performed a hypergeometric test and determined that there was in fact significant enrichment of previously described transit-amplifying genes in the 'TAC-TEC' population (p=$1.44\times10^{-34}$; *Figure 2a* and *Figure 2—figure supplement 1b*). Transit-amplifying cells are defined by rapid cycling (*Rangel-Huerta and Maldonado, 2017*; *Zhang and Hsu, 2017*), so we hypothesized that the 'TAC-TEC' cluster would contain cycling TECs. To test this we used cyclone (*Scialdone et al., 2015*), an algorithm that determines cell-cycle state based on expression of cell-cycle genes, which revealed that the majority (104/123) of cells in the G2/M phase were located in the 'TAC-TEC' cluster (*Figure 2b*). Notably, the mapping of cycling cells to the 'TAC-TEC' population provides support for the RNA velocity prediction that 'TAC-TECs' populate other major mTEC subsets (*Figure 1E*). Expression of *Hmgb2* specifically correlated with cells being classified as G2/M and many cycling cells also expressed *Fezf2* (*Figure 2b*). Many cycling 'TAC-TECs' expressed *Aire* and *Ccl21a* but they were often not co-expressed in the same cells, consistent with heterogeneity described in other transit-amplifying populations (*Zhang and Hsu, 2017*; *Figure 2b* and *Figure 2—figure supplement 1c*).

To determine if the cycling population could be detected at the protein level, we performed intracellular flow cytometry for Ki67. For this analysis, MHCII and Aire gates were drawn at the lower boundry of the high-density contours to separate bona fide, mature MHCII[hi] and Aire[hi] populations from all others (negative-intermediate). Intracellular staining revealed a Ki67[bright] TEC population that was MHCII[int] and Aire[−/low], implying that cell division immediately precedes high levels of *Aire* expression in a subset of cells (*Figure 2c*). However, 'TAC-TECs' were not found to express previously described canonical stem cell markers (*Arnold et al., 2011*; *Lyle et al., 1999*; *Redvers et al., 2006*; *Yu et al., 2006*), underscoring their identity as a transit-amplifying population and distinct from stem-like progenitors (*Figure 2—figure supplement 1d*). These data demonstrate the transcriptional signature of the 'TAC-TEC' population is enriched for transit-amplifying genes and this population is also actively cycling, both consistent with the classification of 'TAC-TECs' as transit-amplifying cells.

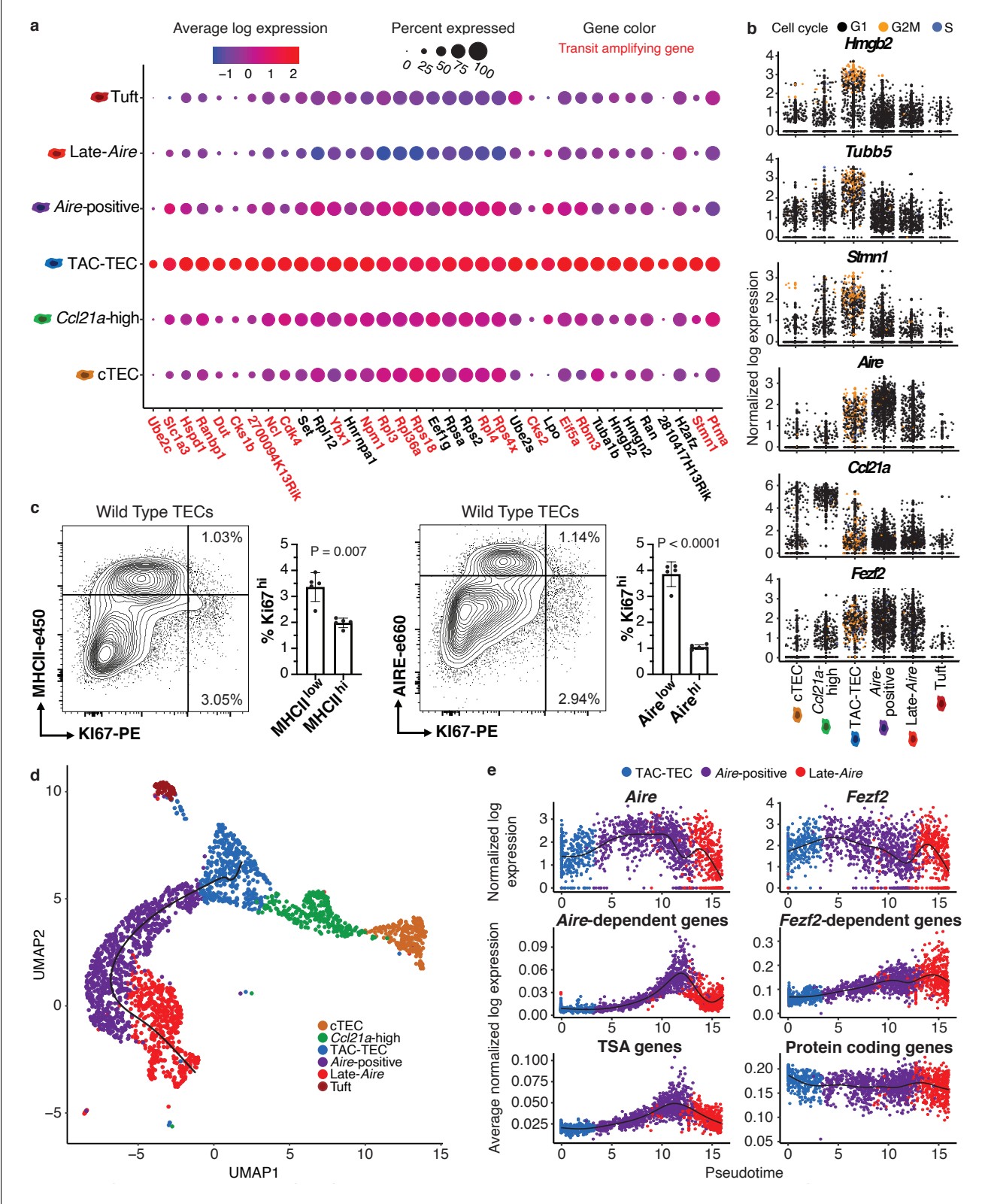

**Figure 2.** Characterization of TAC-TECs and developmental dynamics from this population. (**A**) Summary dot plot of expression of genes differentially upregulated uniquely in the TAC-TEC population (adj-p-val<0.05 and log|fold change| > 2) across mTEC populations. Red text color represents genes previously associated with transit-amplifying cells, DE genes were enriched for transit-amplifying genes (p=1.44×10-34). Color indicates average expression and size indicates percent of cells expressing the transcript. (**B**) Normalized log expression of selected cell-cycle genes, chromatin modifiers,

*Figure 2 continued on next page*

*Figure 2 continued*

and genes important for mTEC function. Colors represent cell-cycle state. (C) Flow-cytometric analysis of mTEC (AIRE and MHCII) vs a cell-cycle (KI67) markers (three pooled thymi per replicate, n = 5 replicates). Bar plots represent percentages of cells positive for Ki67 for each population (data are mean ±s.d. n = 5 mice). (D) Predicted pseudotime line (slingshot algorithm) depicting developmental trajectory exclusively for cells in the Aire branch. (E) Expression patterns of genes (log expression) and gene sets (average log expression) for cells in the Aire branch of development across pseudotime (see E). Color represents population. See also *Figure 2—figure supplements 1* and *2*.

The online version of this article includes the following figure supplement(s) for figure 2:

**Figure supplement 1.** Characterization of TAC-TEC Population.
**Figure supplement 2.** TSAs are expressed late in pseudotime.

## TSAs appear late in mTEC development

A key function of mTECs is to express and display TSAs to developing T cells. While mTECs are capable of expressing the majority of the protein-coding genome, little is known about the kinetics of TSA expression in vivo and how specific regulatory factors impact TSA expression during mTEC maturation. Performing pseudotemporal analysis on single-cell RNA-seq data can be used to predict lineage relationships and the order of events that occurs during development (*Cheng et al., 2018*; *Cohen et al., 2018*; *Loo et al., 2019*). To provide insight into the process of TSA expression, we examined gene expression across predicted developmental time using pseudotime to computationally order cells. To accomplish this, we used the Fantom database to compile a list of TSAs by identifying genes detected in five or fewer tissues (*Brennecke et al., 2015*; *Forrest et al., 2014*). We determined the average expression of all TSAs in each individual cell and plotted cells along pseudotime scores provided by Slingshot (*Street et al., 2018*; *Figure 2d*). Based on pseudotime predictions, TSA expression did not peak until well after the initiation of *Aire* expression and was maintained even after *Aire* expression decreased (*Figure 2d and e*). To determine if the increase of average TSA expression late in pseudotime was because of changes in the number of TSAs expressed or the expression level of TSAs, we observed both the percent of TSAs expressed per cell and the average expression of only the expressed TSAs (*Figure 2—figure supplement 2a and b*). The percent of TSAs expressed peaked late in pseudotime and followed the pattern described for all TSAs (*Figure 2—figure supplement 2a*). In contrast, the average expression of only expressed TSAs did not show this pattern and instead increased gradually throughout pseudotime (*Figure 2— figure supplement 2b*). These results demonstrate that the number of TSAs expressed peaks late in pseudotime rather than the expression level of TSAs and suggest *Aire* expression must be well-established before TSA expression can occur.

To further dissect TSA expression patterns, we used previously described *Aire*- and *Fezf2*-dependent gene sets identified through characterization of the respective knockout animal models (*Sansom et al., 2014*; *Takaba et al., 2015*). Although *Aire* expression was detectable in dividing mTECs (*Figure 2b*), the pattern of *Aire*-dependent gene expression (a slow increase to peak levels followed by a decrease) was very similar to that of all TSAs (*Figure 2e*). *Fezf2* expression peaked earlier in pseudotime than *Aire* expression and, in contrast to *Aire*-dependent gene expression, *Fezf2*-dependent genes were expressed more uniformly across pseudotime (*Figure 2e*). To ensure the timing of gene expression over pseudotime wasn't affected by TSAs, we repeated the analysis after removing TSA genes from the ordering and observed similar timing of events across pseudotime (*Figure 2—figure supplement 2c*). Together, these results provide a model for the kinetics of gene expression across development of *Aire*-positive mTECs and illustrate the differences in expression between *Aire* and *Fezf2*.

## Aire lineage tracing mice demonstrate cellular relationships in the Aire branch of mTEC development

While single-cell transcriptomics can identify subpopulations, widely used tools for describing cellular dynamics rely upon computational assumptions and predictions. Thus, bioinformatics can predict lineage relationships between mTECs but cannot, in isolation, prove those relationships. To provide experimental support for the predicted lineage relationships, we first utilized an *Aire* lineage tracing system in which *Aire*-expressing cells are inducibly and indelibly labeled with the fluorescent reporter protein ZsGreen after tamoxifen treatment (*Aire^{CreERT2};Rosa26^{CAG-stopflox-zsGreen}*) (*Metzger et al.,*

*2013*). This allows for discrimination between cells that express *Aire* or have passed through an *Aire*-expressing state (positive for ZsGreen) and those that have never expressed *Aire* (negative for ZsGreen) (*Figure 3a*). Because transient *Aire* expression occurs during embryogenesis (*Nishikawa et al., 2010*), tamoxifen was used to allow strict temporal control of Cre recombinase activity as previously described (*Metzger et al., 2013*). In this system, mice must be treated with tamoxifen to induce Cre translocation to the nucleus and initiate reporter expression. This event occurs only in cells expressing *Aire* and, therefore, Cre. However, once the Stop codon has been removed labeling is permanent and the reporter will continue to be expressed in cells after *Aire* expression ceases. Following a 10 day tamoxifen label, total TECs were purified by flow cytometry (EpCAM$^+$ CD45$^-$) and subjected to scRNA-seq on the 10x Genomics platform (*Zheng et al., 2017*; *Figure 3a*). A total of 1387 cells were included in the downstream analysis.

Single cells were projected into a reduced-dimensional space using UMAP and were clustered based on the top 12 principle components (*Becht et al., 2018*; *Butler et al., 2018*; *Stuart et al., 2019*; *Figure 3b* and *Figure 3—figure supplement 1a*). First, we used the gene-expression patterns from our initial bioinformatic analysis from control mice to provide putative identities to the clusters observed in the lineage tracing animals (*Figure 3c*). Notably, there was strong agreement between the clusters identified in control mice and those observed in the lineage tracing mice, indicating lineage trace samples were comparable to the control mice (*Figure 3—figure supplement 1*). Next, we examined the robustness of the computational lineage predictions (*Figure 2d*) by comparing the expression pattern of the ZsGreen lineage reporter to *Aire*. Specifically, in the 'Aire-positive' cluster, 92% of cells expressed high levels of both *Aire* and ZsGreen (*Figure 3c and e*, *Figure 3—figure supplement 2*, and *Supplementary file 1*). In the 'Late-*Aire*' cluster, 46% expressed both *Aire* and ZsGreen, while 43% expressed only ZsGreen, confirming prior widespread *Aire* expression (*Figure 3c and e*, *Figure 3—figure supplement 2*, and *Supplementary file 1*). The 'Late-*Aire*' cluster was also marked by a small number of cells (14%) expressing high levels of *Krt10*, which has previously been described as a maker of post-*Aire*, terminally differentiated cornifying mTECs (*Metzger et al., 2013*; *Wang et al., 2012*; *Figure 3c*, *Figure 3—figure supplement 2*, and *Supplementary file 1*). In contrast, 52% of cells in the 'TAC-TEC' cluster expressed both *Aire* and ZsGreen but 18% of cells expressed only *Aire* (*Figure 3c and e*, *Figure 3—figure supplement 2*, and *Supplementary file 1*). This was expected because ZsGreen reporter expression requires multiple downstream events after *Aire* expression, including Cre$^{ERT2}$ translation, nuclear localization in the presence of tamoxifen, Cre-mediated Rosa-Stop-flox excision, and ZsGreen transcription and translation, which delay the kinetics of ZsGreen expression in comparison to *Aire*. This transcriptional lag provides a useful signature (*Aire*$^+$, ZsGreen$^-$) that is indicative of recent *Aire* induction and was the pattern uniquely observed within the 'TAC-TEC' cluster (*Figure 3c and e*, *Figure 3—figure supplement 2*, and *Supplementary file 1*). In contrast, co-expression of ZsGreen and *Aire* in the 'Aire-positive' cluster was nearly ubiquitous and consistent with established *Aire* expression (*Figure 3c and e*, *Figure 3—figure supplement 2*, and *Supplementary file 1*). There were also 34% of cells that highly expressed ZsGreen in the '*Ccl21a*-high' population but only 2% of these cells also expressed *Aire* (*Figure 3d,e*, and *Supplementary file 1*). To determine if ZsGreen was expressed in the '*Ccl21a*-high' population at the protein level, we performed flow cytometry on *Aire* lineage tracing mice treated with tamoxifen as above and stained for intracellular CCL21 protein. Approximately half of cells that expressed CCL21 also expressed ZsGreen reporter but 25% of these double-positive cells were MHCII$^{lo}$, indicating that a subset of '*Ccl21a*-high' cells are likely downstream of an *Aire*-expressing population (*Figure 3f* and *Figure 3—figure supplement 3*). The finding that *Ccl21a* cells are marked by ZsGreen is consistent with the pseudotime prediction that the '*Ccl21a*-high' population is downstream of the 'TAC-TEC' population rather than the progenitor of the 'Aire-positive' population. Taken together, these data are consistent with the pseudotime predictions and provide further support for the 'TAC-TEC' cluster as the transit-amplifying population feeding the 'Aire-positive' and '*Ccl21a*-high' compartments (*Figure 3c–f*, *Figure 3—figure supplements 2* and *3*, and *Supplementary file 1*).

Notably, we again observed heterogeneity in the 'TAC-TEC' cluster with some cells expressing *Aire* and other cells expressing *Ccl21a*, but rarely high levels of both (*Figure 3e*). This again demonstrates that the 'TAC-TEC' population is a heterogeneous population with some cells more closely resembling the 'Aire-positive' population and other cells more closely resembling the '*Ccl21a*-high' population. The combination of bioinformatics with lineage tracing enabled the identification of a

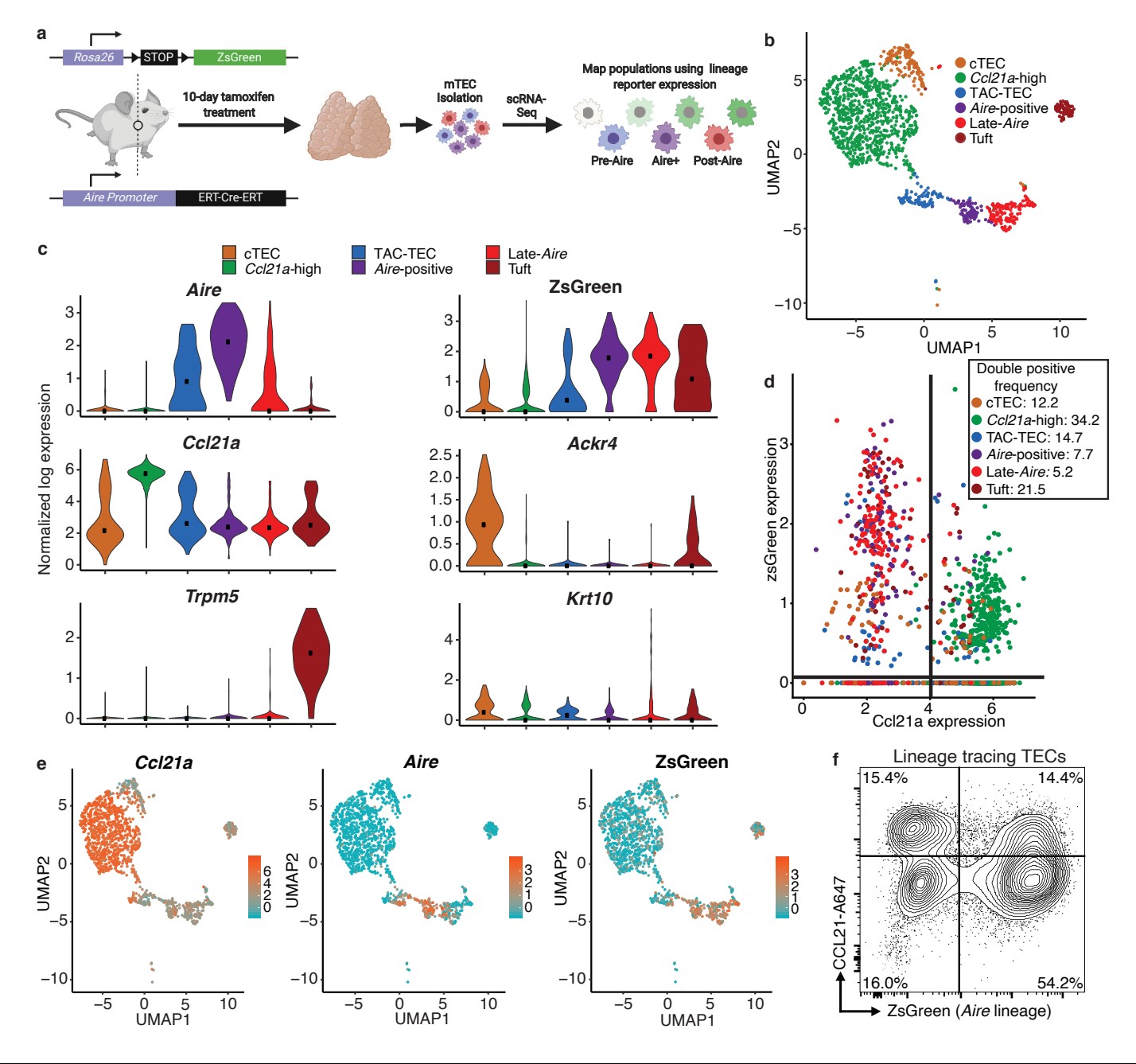

**Figure 3.** Aire–lineage tracing identifies immediate precursors of *Aire*-positive cells. (A) Outline of Aire–lineage trace experiment. Aire–lineage tracing mice were treated with Tamoxifen for 10 days. Thymi were then isolated, FACS-purified, and subjected to single-cell RNA-seq using 10x Genomics. In lineage trace mice, any cell that has ever expressed *Aire* will express ZsGreen. (B) UMAP visualization of all 1387 cells and clustering (color) of cells from the *Aire* lineage tracing mouse (three pooled thymi). Each point represents a cell. (C) Normalized log expression of marker genes for known mTEC populations across cells in each cluster. Color represents cluster, black dot is median expression. (D) zsGreen expression vs *Ccl21a* expression across all mTEC populations. Lines indicate values used to determine positive expresssion (0 for zsGreen, four for *Ccl21a*). Color represents population, axes are log-normalized expression. The frequency of double positive cells within each population is indicated in the key. Frequencies for the other three quadrants can be found in **Supplementary file 1**. (E) UMAP dimensional reduction visualizations of the *Aire* lineage tracing cells. Color represents normalized log expression of *Ccl21a, Aire*, and ZsGreen for mTEC populations. Each point represents a cell. (F) Flow-cytometric analysis of intracellular CCL21 staining vs the Aire–lineage marker ZsGreen (three pooled thymi). See also **Figure 3—figure supplements 1**, **2** and **3**.

The online version of this article includes the following figure supplement(s) for figure 3:

**Figure supplement 1.** Classification of *Aire* lineage tracing cells.

**Figure supplement 2.** Characterization of ZsGreen in mTEC populations.

*Figure 3 continued on next page*

*Figure 3 continued*

**Figure supplement 3.** Classification of the CCL21 population by flow cytometry.

transit-amplifying population that precedes *Aire*-expressing mTECs and provided a high-resolution map of *Aire*-adjacent TEC subsets for further analysis.

## In vivo RANKL blockade reveals ordering of the Aire branch of mTEC development

To provide additional, independent support for the mTEC lineage relationships inferred by our analysis of unperturbed thymus, we employed a model of transient mTEC ablation in which RANK signaling was blocked in vivo using anti-RANKL antibodies (*Khan et al., 2014*; *Metzger et al., 2013*). We have previously characterized this system and its kinetics in detail (*Khan et al., 2014*). Notably, by flow cytometry, there is a pronounced decrease in the absolute number of AIRE⁺ and MHCII^hi mTECs following antibody blockade, while MHCII^lo mTEC numbers are less impacted (*Khan et al., 2014*). Structurally, corticomedullary architecture is maintained and KRT5 remains easily detectable by immunofluorescence but AIRE staining is transiently absent (*Khan et al., 2014*). Because *Aire*-expressing mTECs are slowly repopulated following treatment with anti-RANKL, we used this model to permit the observation of developmental progression of adult mTECs during recovery from acute, transient ablation.

Wild-type mice were treated with anti-RANKL blocking antibody and TECs were isolated for scRNA-seq using the 10x Genomics platform (*Zheng et al., 2017*) over a 10 week time course (*Figure 4a*). A total of 8453 cells were included in the downstream analysis. After initial processing, the ablation samples, control samples, and *Aire* lineage tracing sample were combined by Seurat's canonical correlation analysis, using the top 1000 most highly variable genes from each sample to account for batch effects (*Butler et al., 2018*). These canonical correlations were used to perform dimensionality reduction and clustering on all combined samples (*Figure 4—figure supplement 1a*).

To determine how cell populations changed following treatment with anti-RANKL, we inferred population labels for clusters from the combined experiment based on the identity of the *Aire* lineage tracing cells contained in the respective clusters (*Figure 3c* and *Figure 4b*). For example, if cluster 1 contained *Aire* lineage tracing cells classified as 'Aire-positive', cluster 1 would be classified using this label. We found that gene-expression patterns between populations closely resembled those seen in the *Aire* lineage tracing experiment (*Figure 4—figure supplements 1*, *2* and *3*). As expected, anti-RANKL treatment decreased the proportion of all *Aire*-expressing and post-*Aire*-expressing populations throughout the time course of treatment and recovery (*Figure 4c* and *Figure 4—figure supplement 1b*). The decrease in all mTECs, and specifically AIRE⁺ mTECs (both in absolute count and relative frequency) was also observed by intracellular flow cytometry (*Figure 4—figure supplement 1d*). Although the *Ccl21a* population appeared to expand following treatment with anti-RANKL, intracellular flow cytometry revealed no significant difference in absolute counts of CCL21⁺AIRE⁻ cells at week four in treated and control mice (*Figure 4d* and *Figure 4—figure supplement 2b*). Thus, only the *Aire*-expressing developmental branch was depleted by treatment with anti-RANKL.

If the 'TAC-TEC' population precedes the 'Aire-positive' and 'Late-Aire' populations in development as predicted by single-cell velocity and *Aire*-lineage tracing, recovery should follow this chronological order after ablation. To test this hypothesis, we overlaid the predicted pseudotemporal ordering of the *Aire*-expressing branch from all samples generated by Slingshot (*Street et al., 2018*) with the RANKL time course (*Figure 4e*). Two weeks following treatment with anti-RANKL, the majority of remaining cells were located in the 'Late-Aire' cluster (*Figure 4f*). These 'Late-Aire' cells likely represented the final wave of *Aire*-expressing cells immediately before RANKL blockade, consistent with estimates of mTEC half-life (*Gray et al., 2006*; *Figure 4f*). Four weeks after treatment, cells at the earliest point in pseudotime began to return, while the number of cells late in pseudotime decreased (*Figure 4f*). Six weeks into recovery, the cell populations recovering continued to shift further along the pseudotime axis, and by week 10, the cell distribution over pseudotime closely resembled that of untreated controls (*Figure 4f*) in agreement with immunostaining showing that anti-RANKL thymi at week 10 were indistinguishable from control thymi (*Khan et al., 2014*). The

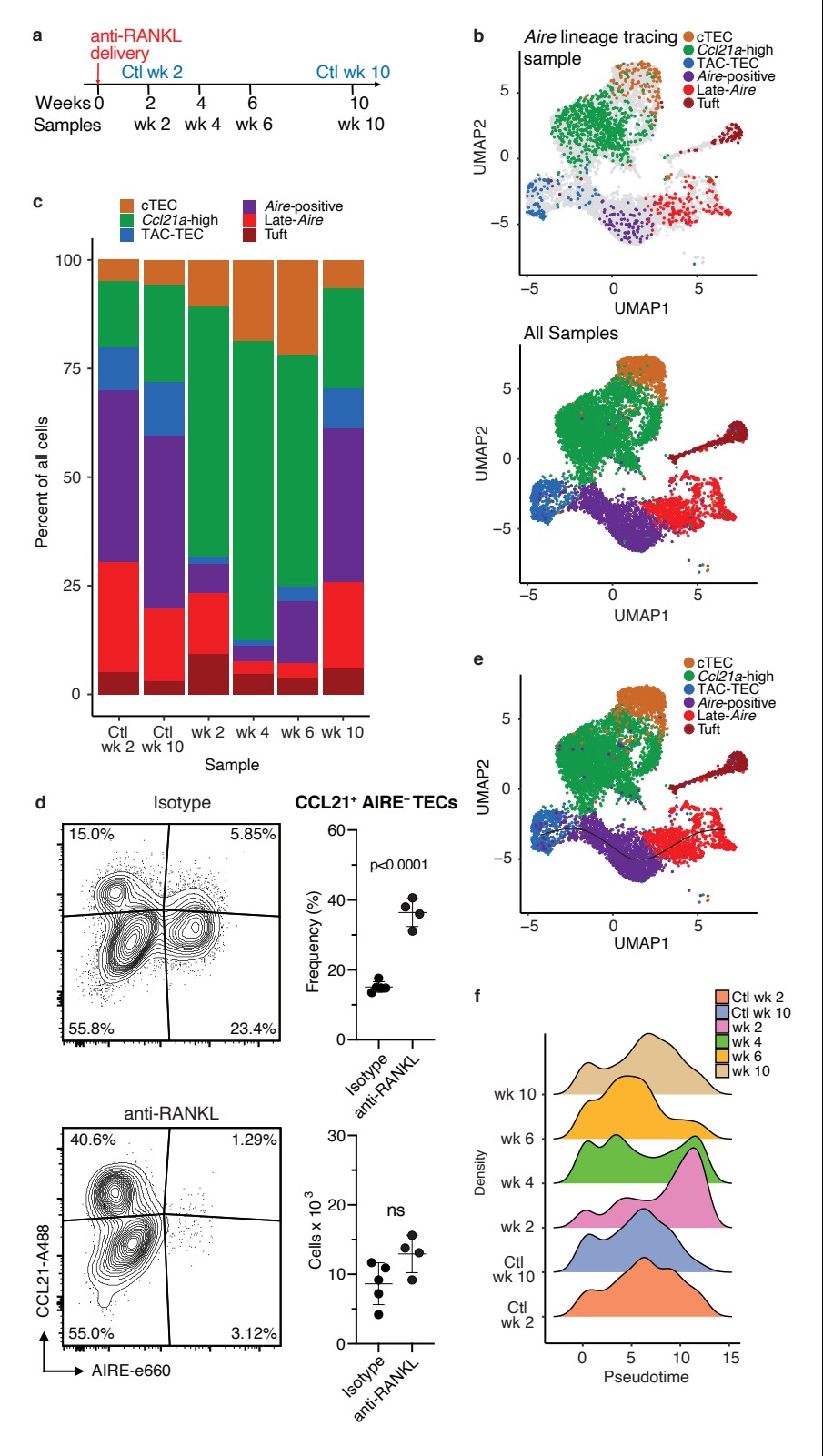

**Figure 4.** Treatment with anti-RANK ligand decreases the relative size of the entire Aire-expressing mTEC population. (**A**) Overview of the experimental protocol. Anti-RANKL was given over the course of a week, and thymi were sequenced at weeks 2, 4, 6, and 10 following treatment. Isotype control thymi were also sequenced at weeks 2 and 10. 3 thymi were pooled for all samples. (**B**) UMAP projections of all 8453 cells from all samples. Top: color represents original *Aire* trace identity (see ***Figure 1***), gray represents all other samples. Bottom: color represents inferred population labels. Identity of

*Figure 4 continued*

cells in all samples was inferred from the original identity of *Aire* lineage tracing cells. (C) Proportions of cells in each population for each sample. Color is cell population. (D) Flow-cytometric analysis of the Ccl21a population at week four following treatment with anti-RANKL and an age-matched isotype control mouse (two pooled thymi per replicate, n = 4 replicates for treatment, n = 5 for controls). Plots show the proportion of all TECs in the CCL21 population and the absolute number of cells in the CCL21 population for the anti-RANKL treated and isotype control samples (data are mean ± s.d.). (E) Predicted pseudotime line (slingshot algorithm) depicting developmental trajectory for cells from all samples in exclusively the *Aire* branch. (F) Density plots of cells in the *Aire* branch of development across pseudotime (see E) for each sample. Color represents sample. See also *Figure 4— figure supplements 1*, *2* and *3*.

The online version of this article includes the following figure supplement(s) for figure 4:

**Figure supplement 1.** Classification of cell populations following treatment with anti- RANKL.
**Figure supplement 2.** Classification of cells following anti-RANKL treatment.
**Figure supplement 3.** Gene-expression patterns across experiments.

close correlation of the in vivo recovery of the *Aire*-developmental branch and the ordering predicted by pseudotime provides strong validation of the pseudotime bioinformatic model. Finally, comparison of gene expression between the final timepoint and isotype control treated animals confirmed mTECs had fully recovered by week 10 (*Figure 4—figure supplement 2c*). Thus, the pattern of mTEC recovery after transient anti-RANKL ablation provides in vivo support for the predictions made based on lineage tracing and bioinformatic approaches.

## The TAC-TEC population contains cells primed for both the Aire and Ccl21a fates

Our data suggest the TAC-TEC population is a heterogeneous population consisting of a *Ccl21a*-expressing and *Aire*-expressing population. To further explore the heterogeneity of this population, we examined the cells that remain in the 'TAC-TEC' population following anti-RANKL antibody treatment. Although the 'TAC-TEC' population was partially depleted after treatment with anti-RANKL, a small number of 'TAC-TECs' remained. Intracellular flow cytometry revealed that there were fewer Ki67$^+$ cycling cells (p<0.0005; *Figure 5a*). Ki67$^+$ cells were nearly entirely depleted in the AIRE-hi population (p<0.0006), a 20-fold reduction, but there was only a 3.5-fold reduction in the AIRE-lo population (p<0.001) suggesting that the TAC-TEC population is not uniformly affected during anti-RANKL treatment (*Figure 5a*). To better understand the 'TAC-TECs' that persisted during antibody treatment, we performed focused analysis on all 'TAC-TECs' collected in the study (n = 511 cells; *Figure 5b*). The persisting 'TAC-TEC' population displayed the same proportion of cells in G2/M and showed similar expression patterns of chromatin-modifying factors (*Figure 5c* and *Figure 5— figure supplement 1a and b*). Across the timecourse, the proportion of *Ccl21a*-expressing 'TAC-TECs' was increased at the expense of *Aire*-expressing 'TAC-TECs' at two and four weeks following treatment but frequencies were beginning to normalize by week 6 (*Figure 5d* and *Figure 5—figure supplement 1c*). Although the expression of *Ccl21a* and *Aire* changed throughout the timecourse, the expression of *Fezf2* was less significantly affected by treatment with anti-RANKL (*Figure 5d* and *Figure 5—figure supplement 1a*). While the proportion of *Ccl21a* expressing cycling cells increased following treatment with anti-RANKL, the number of cells in the 'TAC-TEC' population that expressed only *Ccl21a* remained similar throughout the timecourse (*Figure 5—figure supplement 1d*). The consistency observed in the *Ccl21a*-expressing cycling population following treatment with anti-RANKL agrees well with our finding that the absolute number of '*Ccl21a*-high' mTECs did not change dramatically over the timecourse and shows that *Aire*-expressing and *Ccl21a*-expressing 'TAC-TECs' are not equally affected by anti-RANKL antibody treatment.

To better understand population dynamics within the 'TAC-TEC' cluster, we repeated canonical correlation analysis between all samples using only these cells. Dimensionality reduction and clustering revealed three sub-populations (*Figure 5e*). To understand the cells within each cluster, we performed differential expression analysis between the three clusters (*Figure 5—figure supplement 2*). Many of the genes differentially upregulated in cluster 1, such as *Calcb, Aire, H2-ab1*, were also expressed in the '*Aire*-positive' population (*Figure 5—figure supplement 2*). The genes upregulated in cluster 0, such as *S100a14, Calcb, Cdx1*, and *Aire* were also expressed in the '*Aire*-positive' population (*Figure 5—figure supplement 2*). The genes upregulated in cluster 2, including *Ccl21a, Krt17, Krt5, and Id3*, were also most highly expressed in the '*Ccl21a*-high' population (*Figure 5—*

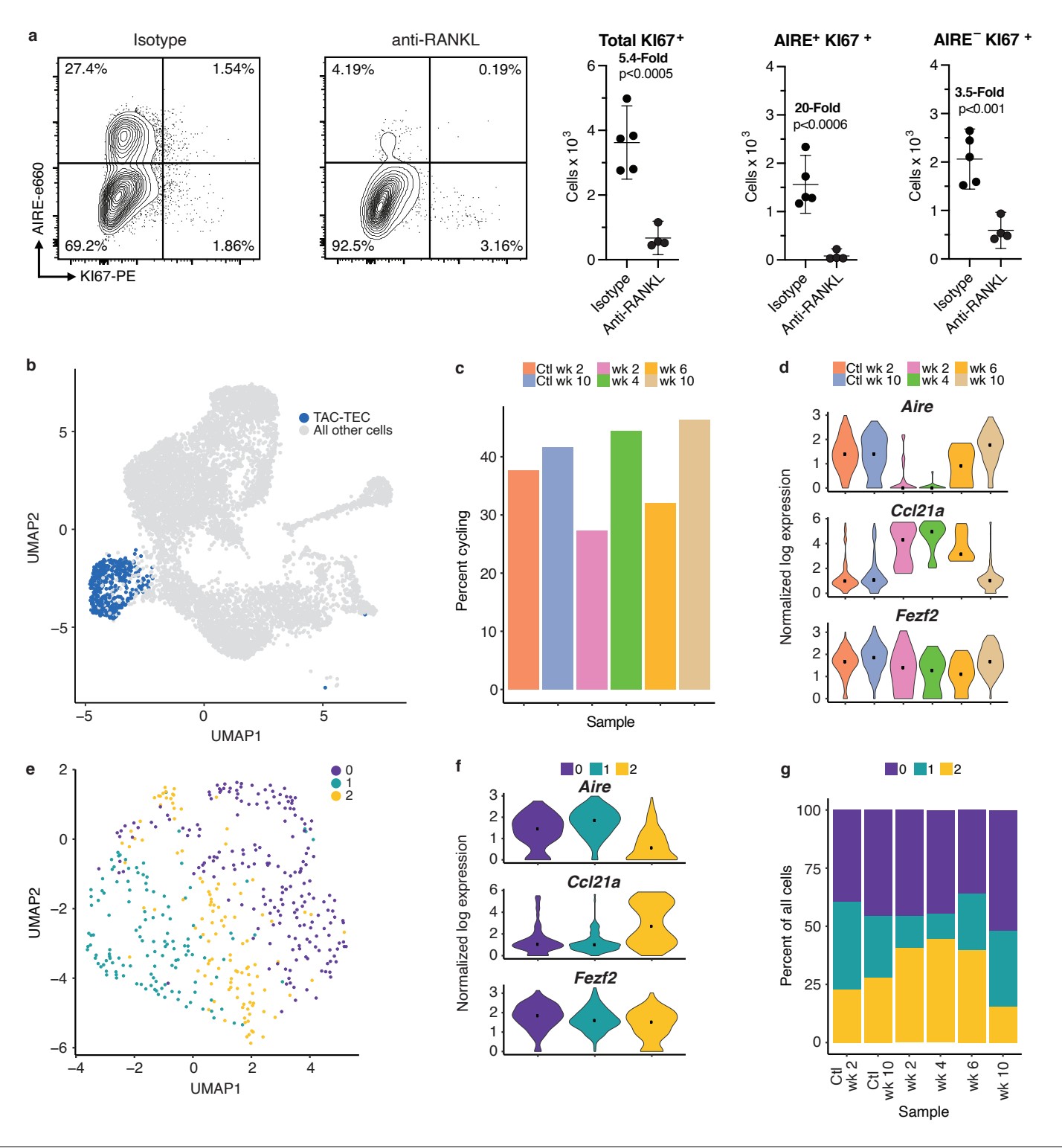

**Figure 5.** Analysis of the TAC-TEC population provides insights into population dynamics. (**A**) Flow-cytometric analysis of KI67-expressing cells at week four following treatment with anti-RANKL and an age-matched isotype control mouse (two pooled thymi per replicate, n = 4 replicates for treatment, n = 5 for controls). Plots show absolute number of cells that are KI67+, KI67+AIRE+, and KI67+AIRE- in anti-RANKL treated and isotype control samples (data are mean ± s.d.). (**B**) Analysis for this figure was restricted to the persistent TAC-TEC population of all samples (511 total cells), highlighted on the UMAP of all cells. (**C**) Percentage of cycling cells within the TAC-TEC population, colored by sample. (**D**) Normalized log expression of *Aire*, *Ccl21a*, and *Fezf2*, across TAC-TECs in each sample. Color represents sample, black dot is median expression. (**E**) UMAP visualization and clustering of TAC-TEC

*Figure 5 continued on next page*

*Figure 5 continued*

reanalysis. Each point represents a cell in the TAC-TEC population, color represents new cluster identity (labeled 0, 1, or 2). (F) Normalized log expression of *Aire, Ccl21a,* and *Fezf2* in the TAC-TEC population across cells in each cluster. Color represents sample, black dot is median expression. (G) Proportions of cells in each cluster in each sample. Color represents cluster. See also *Figure 5—figure supplements 1* and *2*.

The online version of this article includes the following figure supplement(s) for figure 5:

**Figure supplement 1.** Analysis of the TAC-TEC population.
**Figure supplement 2.** Differentially expressed genes across the 'TAC-TEC' clusters.

*figure supplement 2*). The frequency of cycling cells and levels of chromatin remodeling factors, including *Hmgb2, Hmgn2,* and *H2afx,* and expression levels of *Fezf2* were similar between the three subsets (*Figure 5—figure supplement 1e and f*). However, clusters 0 and 1 were found to have the highest levels of *Aire* and lowest levels of *Ccl21a,* whereas cluster 2 was found to have the lowest levels of *Aire* and highest levels of *Ccl21a,* suggesting an early stage of lineage commitment (*Figure 5f*). Relative to other clusters within each treatment timepoint, the proportion of cells in cluster 2 increased at weeks 2 and 4 after treatment, consistent with analysis of *Ccl21a-* and *Aire*-expressing cells (*Figure 5d and g*). By 6 weeks after treatment the abundance of *Aire*-expressing cluster 1 cells had largely normalized (*Figure 5g*). Notably, despite changes in frequency, the transcriptomic profile of each 'TAC-TEC' cluster remained consistent after RANKL blockade (*Figure 5g*). Taken together, while the 'TAC-TEC' cluster is defined by cycling cells and expression of chromatin remodeling factors, it can be further subsetted to reveal *Aire* or *Ccl21a* primed states. These results demonstrate that the sub-populations are present in all samples, indicating that *Aire-* or *Ccl21a*-primed cells are present in control thymi and treated thymi. Further, these results demonstrate that the *Aire*-primed cells are selectively ablated following treatment with anti-RANKL.

## Aire-dependent TSAs are expressed late in mTEC development and are uniquely sensitive to RANK blockade

Next, we assessed the dynamics of gene expression within the total *Aire*-positive population following anti-RANKL treatment (*Figure 6a*). While the expression of housekeeping genes, including *Gapdh,* and the expression of *Fezf2* did not change following anti-RANKL treatment, *Aire* expression was markedly reduced and with the lowest point at week four followed by recovery (*Figure 6b* and *Figure 6—figure supplements 1b* and *2a*). The recovery of *Aire* expression at week six was consistent with the pseudotime prediction that *Aire* expression peaks early in mTEC development (*Figure 2e*).

To explore the recovery of TSAs, we determined the average expression of genes in each of the gene lists described above (*Aire*-dependent, *Fezf2*-dependent and total TSAs) throughout recovery. Although the expression of all non-TSA genes (all genes, excluding these three gene lists) did not change across the timecourse, average expression of total TSAs and especially *Aire*-dependent genes decreased (*Figure 6c* and *Figure 6—figure supplement 2b*). TSAs and *Aire*-dependent gene expression began to recover at week six but did not fully recover until week 10 (*Figure 6c*). The pronounced delay between the recovery of *Aire*-expressing cells and *Aire*-dependent TSA expression was surprising. However, it supports the bioinformatic prediction that *Aire* expression must be well-established before the expression of TSAs.

The expression of TSA and *Aire*-dependent genes could have decreased because the expression of each gene decreased, fewer total genes were expressed within each group, or fewer cells were present within the *Aire*-positive population. Therefore, we determined how many genes from each list were observed throughout the time course. Before making this calculation, we normalized the number of unique molecular identifiers (UMIs) in each sample to avoid the influence of gene dropout rates (*Figure 6—figure supplement 2c–f*). The cumulative fraction of expressed total TSA and *Aire*-dependent genes decreased from over 75% in untreated controls to less than 20% following ablation, with the lowest fractional expression at week 4 (*Figure 6d* and *Figure 6—figure supplement 3b*). Although *Aire* expression was mostly recovered by week 6, *Aire*-dependent gene expression and TSA expression remained below 25% (*Figure 6d* and *Figure 6—figure supplement 3b*). The average expression of only TSAs detected at week four following ablation did not change dramatically following treatment (*Figure 6—figure supplement 3c*). The number of *Fezf2*-dependent genes

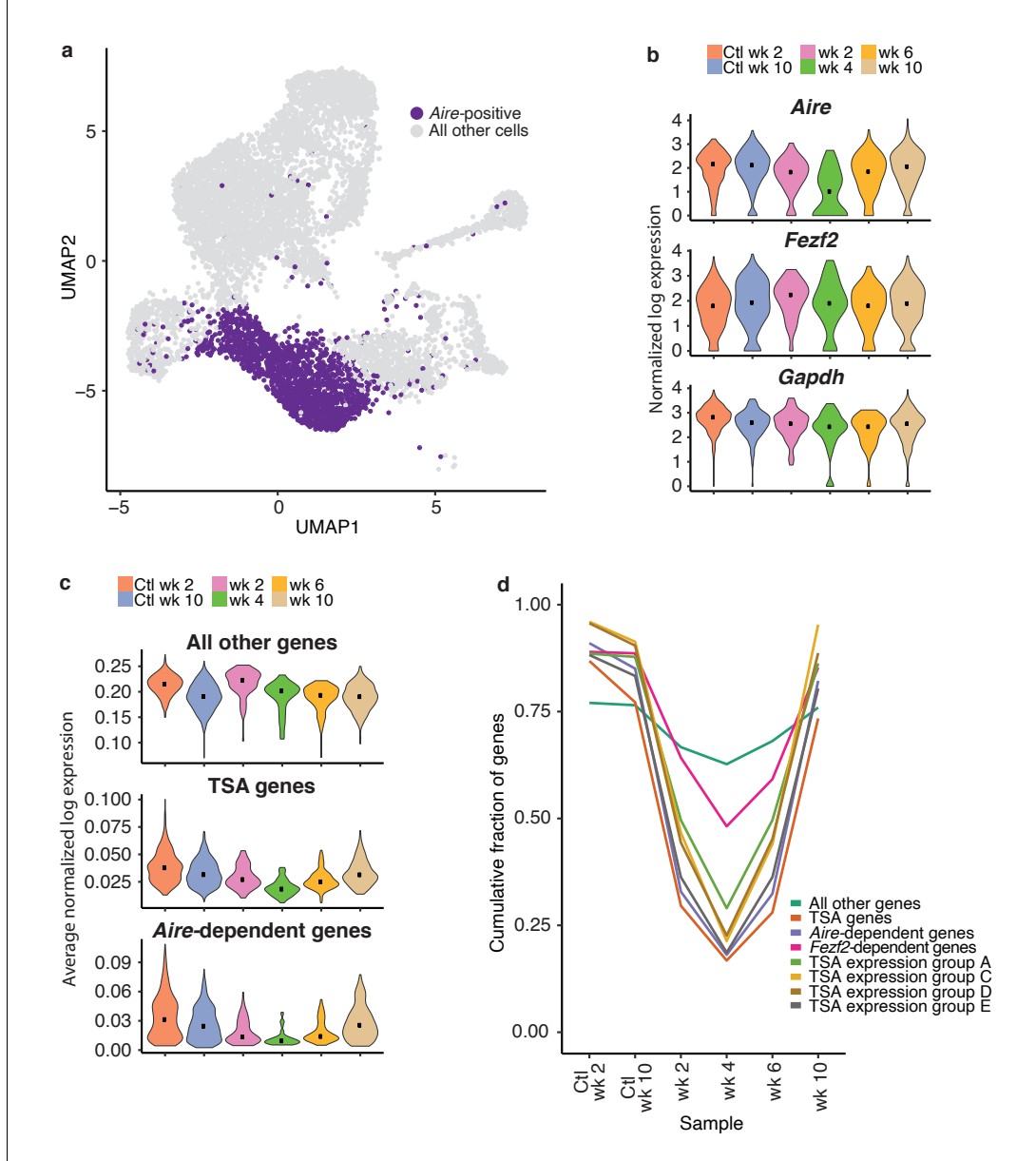

**Figure 6.** TSAs are lost after treatment and recover with *Aire* expression. (A) Analysis for this figure was restricted to the *Aire-positive* population of all samples highlighted on the UMAP of all cells. (B) Normalized log expression of *Aire*, *Ccl21a*, and *Fezf2*, across Aire-positive cells in each sample. Color represents sample, black dot is median expression. (C) Average normalized log expression across gene sets in Aire-positive cells for each sample. Color represents sample, black dot is median expression. (D) Cumulative fraction of genes detected in each sample. Color represents gene set. See also *Figure 6—figure supplements 1*, *2* and *3*.

The online version of this article includes the following figure supplement(s) for figure 6:

**Figure supplement 1.** Expression patterns of *Aire* and *Fezf2* following anti-RANKL treatment.

**Figure supplement 2.** TSA expression occurs after *Aire* expression.

**Figure supplement 3.** Downsampling of genes and cells shows recovery patterns consistent with non-downsampled conclusions.

also decreased from over 80% to 50% (*Figure 6d* and *Figure 6—figure supplement 3b*). Based on this analysis, expression of TSAs and *Aire*-dependent genes decreased because fewer genes from each category are expressed at week four than in the controls.

TSAs in mTECs were previously shown to be co-expressed as gene sets within individual cells in a somewhat ordered pattern (*Brennecke et al., 2015*). To determine if co-expressed TSA gene sets were expressed at different times during mTEC development, we investigated the recovery of a

subset of co-expressed gene sets defined by *Brennecke et al., 2015*. We determined the cumulative fraction of genes in each of the co-expression groups that are expressed throughout recovery following treatment with anti-RANKL (*Figure 6d*). All co-expression gene groups had similar patterns to the *Aire*-dependent genes and all TSAs. For all co-expression groups, we detected the fewest genes at week four following treatment (*Figure 6d*). Genes from the co-expression groups began to recover by week 6, had fully recovered by week 10, and all co-expression groups appeared to recover with similar kinetics (*Figure 6d*).

To determine if fewer genes were detected because the 'Aire-positive' population contained fewer cells following anti-RANKL treatment, we down-sampled cells from the week 10 control to the number of cells observed in each of the recovery samples. After down-sampling to equivalent cell numbers, we still observed fewer *Aire*-dependent and total TSA genes expressed after treatment (*Figure 6—figure supplement 3d*). The total number of genes expressed per cell did not change over the course of recovery although the number of total TSA genes per cell, and especially *Aire*-dependent genes, did decrease (*Figure 6—figure supplement 3d*). Overall, the gene-expression patterns observed during recovery of mTECs mirrored the gene-expression dynamics predicted by pseudotime.

## Discussion

We integrated single-cell RNA-sequencing with lineage tracing and transient in vivo ablation to establish a high-resolution map of events that occur during mTEC development. While other single-cell studies have identified similar sub-populations of mTECs, our approach has enabled, for the first time, the identification and validation of lineage relationships between these populations across the Aire branch of mTEC development. Critically, our conclusions extend beyond traditional bioinformatic analysis of high-dimensional single-cell data by including in vivo lineage reporters and an in vivo transient ablation model. In this way, we revealed a transit-amplifying population of mTECs which we dubbed 'TAC-TECs'. We provide both experimental and computational evidence that 'TAC-TECs' are the immediate precursor of *Aire*-expressing MHCII[hi] mTECs. The 'TAC-TECs' have a differentiating phenotype, including high expression of histone proteins, chromatin-modifying factors, and cell-cycle–related genes. The preponderance of factors associated with chromatin organization within 'TAC-TECs' provides transcriptional evidence that chromatin remodeling may occur immediately before high levels of *Aire* expression, and this temporal relationship may establish a permissive chromatin landscape upon which the Aire complex can act. The mechanisms underlying the extraordinary ability of mTEC-hi to heterogeneously express TSAs covering the majority of the protein-coding genome remain poorly understood. The observation of this transient state immediately preceding accumulation of high levels of the Aire complex may aid future studies directed at understanding how Aire accesses and interacts with the genome.

The identification of a transit-amplifying population immediately preceding the *Aire*-expressing mTEC-hi population is consistent with previous reports interrogating mTEC development. *Gäbler et al., 2007* used BrdU labeling to demonstrate that cells from both mTEC-hi and mTEC-lo populations were post-mitotic, but a higher percentage (80%) of mTEC-hi cells were post-mitotic than mTEC-lo (30%). This is consistent with our observation that there is a higher frequency of cycling cells expressing *Aire* than cycling cells expressing high levels of *Ccl21a* and a higher rate of ZsGreen expression in the 'Late-Aire' branch than the 'Ccl21a-high' branch. Gray et al. also showed that *Aire*-positive mTECs incorporated BrdU, consistent with our model that these cells derive from a cycling precursor (*Gray et al., 2007*). By flow cytometry, they identified a cycling population that they suggest may be a transit-amplifying population based on cell-cycle dynamics alone (*Gray et al., 2007*). The rapid turnover of *Aire*-expressing mTECs and their ability to recover from acute injury, such as treatment with anti-RANKL, further suggest a rapidly cycling precursor must be populating this compartment. While previous studies have suggested the presence of transit-amplifying cells in the medulla and that *Aire*-expressing mTECs follow a cycling population, we confirm the existence of a transit-amplifying population using defined gene-expression patterns and uniquely demonstrate that this transit-amplifying population immediately precedes *Aire* expression by leveraging both lineage tracing and transient ablation. Additionally, the high resolution of our single-cell data provides a detailed road-map of gene-expression events from the transit-amplifying population through the expression of TSAs.

Previous studies have suggested *Ccl21a*-expressing cells give rise to *Aire*-expressing mTECs (*Onder et al., 2015*). However, disruption of the *Ccl21a*-expressing population has been shown to have minimal impact on the *Aire*-expressing compartment, arguing against a precursor–product relationship (*Kozai et al., 2017*; *Lkhagvasuren et al., 2013*). Furthermore, Gray et al. demonstrated that mTEC-lo can give rise to *Aire*-expressing mTEC-hi with low efficiency, suggesting only a minor subset of mTEC-lo maintain this potential (*Gray et al., 2007*). In the context of our findings, one possible explanation is that the mTEC-lo fraction included infrequent 'TAC-TECs' amongst the majority 'Cc21a-High' mTEC-lo. We therefore speculate that the mTEC-lo cells described by Gray et al., which eventually give rise to *Aire*-expressing mTECs, are in fact proliferating 'TAC-TECs'.

We provide multiple lines of evidence that we believe support a model in which *Ccl21a*-expressing cells are not the progenitors of the *Aire*-expressing mTECs but are a distinct population (*Lkhagvasuren et al., 2013*). First, single-cell velocity predicted the '*Ccl21a*-high' cluster was a terminal population and identified 'TAC-TECs' as the precursor to '*Aire*-positive' mTECs. Second, cycling cells were confined to the 'TAC-TECs' and were not seen elsewhere in the thymic epithelial cells examined. Third, the 'TAC-TEC' population was heterogeneous, with some cells expressing *Ccl21a* and other cells expressing *Aire,* consistent with lineage-biased transit-amplifying cells (*Zhang and Hsu, 2017*). Fourth, half the cells in the CCL21 compartment express ZsGreen. This provides experimental evidence that a portion of the *Ccl21a* population is downstream of an *Aire*-expressing progenitor. Because most of the cells in the TAC-TEC population don't express ZsGreen, this population cannot be downstream of the lineage-traced *Ccl21a* cells and therefore *Ccl21a* cells are unlikely the progenitors of the *Aire*-positive population. This finding is most consistent with the branching model proposed by this study. Fifth, treatment with anti-RANKL decreased the frequency of *Aire*-expressing but not *Ccl21a*-expressing cycling 'TAC-TECs'. Because cycling *Ccl21a* cells persisted after ablation, absolute counts of '*Ccl21a*-high' cells were not significantly affected by treatment with anti-RANKL while the '*Aire*-positive' subset was almost completely lost. Indeed, the observation that the 'TAC-TEC' cluster can be subsetted into cycling *Ccl21a*-expressing and *Aire*-expressing populations provides evidence that these transient amplifying cells are primed in a partially RANK-dependent manner to further differentiate in response to continued exposure to inductive signals. Taken together, we believe these results suggest a branching, rather than linear, model for mTEC development and raise exciting new questions about instructive signals other than RANK which are integrated at this critical bifurcation point.

Other recent single-cell studies *Dhalla et al., 2020* have also identified a proliferating mTEC population that is very similar to the 'TAC-TEC' population presented here. *Dhalla et al., 2020* used multiple pseudotime methods, including RNA velocity as we performed here, to determine lineage relationships between mTECs. Using diffusion pseudotime, the authors identified similar trajectories to our analysis but concluded that proliferating mTECs were downstream of the *Ccl21a*-expressing mTECs based on their RNA velocity data (*Dhalla et al., 2020*). While we combine multiple experimental techniques to support the branching model proposed here, aspects of both models could fit our data. For example, the *Ccl21a* population may contain a quiescent stem cell that transitions into a 'TAC-TEC' before becoming an *Aire*-expressing cell. In this scenario, it is possible that cells are blocked from transitioning from the '*Ccl21a*-high' fate into the '*Aire*-positive' fate in the presence of anti-RANKL leading to our observation of only *Ccl21a*-expressing cells in the 'TAC-TEC' population following treatment. Additionally, zsGreen-expressing cells in the 'C*cl21a*-high' population could be cells that are beginning to turn on *Aire* as they transition into an *Aire*-expressing state. Further validation including lineage tracing using methods such as SCAR-trace (*Raj et al., 2018*; *Spanjaard et al., 2018*) or isolation of the TAC-TEC population for evaluation of their ex-vivo differentiation capacity will be required to confirm which model is correct.

It is now well-established that *Aire* expression in mTECs is dependent on RANK signaling. However, the signals required to induce *Fezf2* in the thymus remain less clear. Both lymphotoxin β receptor (LTβR) (*Takaba et al., 2015*) and RANK (*Cosway et al., 2017*) signaling have been implicated by previous studies. While there is evidence that *Fezf2*-expression is RANK-inducible (*Cosway et al., 2017*), here we show that in adult thymus, *Fezf2* expression is generally independent of RANK. Notably, induction of *Fezf2* preceded that of *Aire*, *Fezf2* was expressed broadly within the 'TAC-TEC' cluster, and *Fezf2* expression was largely unaffected by anti-RANKL antibody treatment. Curiously, while *Aire*-dependent gene expression peaked at the transition from '*Aire*-positive' to 'Late-Aire', expression of *Fezf2*-dependent genes increased evenly over pseudotime. In sum, these data

suggest that *Fezf2* is regulated differently than *Aire,* eliminate RANK signaling as a necessary driver of the *Fezf2* TSA axis and support the possible role of LTβ or other alternative regulatory pathways, as suggested (*Takaba et al., 2015*).

Our combination of single-cell sequencing with lineage tracing and recovery from ablation helps defines the lineage relationships between mTECs, identifies the transit-amplifying population that is the precursor of *Aire*-expressing mTECs, and details the functional steps that occur before TSA expression. Our data also suggests that the TAC-TECs may also be the precursor of the 'Ccl21a-high' mTEC population. These data and the high-resolution map of mTEC development will serve as a valuable resource to further dissect the functional steps of mTEC development and, ultimately, how T-cell tolerance and selection is maintained by this heterogenous population of cells. Lastly, this combinatorial approach provides a powerful framework for integrating in vivo reporter tools into transcriptomic datasets and can be applied to future studies aimed at generating high-resolution maps of development of other tissues and cell types.

### Code availability

Plots were made using ggplot2 in R using our personal package that have been made publicly available. The full Snakemake pipeline, our personal analysis package, and scripts to recreate all figures are available on GitHub: https://github.com/kwells4/mtec_analysis (*Wells, 2020*; copy archived at swh:1:rev:d3955fad0d73dc404a93fc2d81b84141c9c79efe) with the companion R package available: https://github.com/kwells4/mtec.10x.pipeline.

## Materials and methods

### Mice

*Aire*<sup>CreERT2</sup>;*Rosa26*<sup>CAG-stopflox-zGgreen</sup> have been previously described (*Miller et al., 2018*). C57BL/6 (Jax #000664) and B6.Cg-Gt(ROSA)26Sortm6(CAG-ZsGreen1)Hze/J (Jax #007906) mice were obtained from The Jackson Laboratory (*Madisen et al., 2010*).

Mice were maintained in the University of California San Francisco (UCSF) specific pathogen-free animal facility in accordance with the guidelines established by the Institutional Animal Care and Use Committee (IACUC) and Laboratory Animal Resource Center and all experimental procedures were approved by the Laboratory Animal Resource Center at UCSF. Age-matched female mice aged 4–14 weeks were used for all experiments unless otherwise specified in the text or figure legends.

### In vivo mouse treatments

For Tamoxifen treatment of mice possessing conditional alleles, Tamoxifen (Sigma-Aldrich) was dissolved in corn oil (Sigma-Aldrich) and 2 mg doses were administered by oral gavage every other day for 10 days with flexible plastic feeding tubes (Instech). Anti-RANKL antibody (Clone IK22/5, BioXCell) or isotype control antibody (clone 2A3, BioXCell) was administered to mice at a dose of 100 ug in PBS every other day via intraperitoneal (i.p.) injections for a total of 3 injections for all experiments.

### Single-cell tissue preparation

Mouse thymi were isolated, cleaned of fat and transferred to DMEM (UCSF Cell Culture Facility) containing 2% FBS (Atlanta Biologics) on ice. For each scRNA-seq sample, three thymi were pooled. Thymi were minced with a razor blade and tissue pieces were moved with a glass Pasteur pipette to 15 ml tubes and vortexed briefly in 10 ml of media. Fragments were allowed to settle before removing the media and replacing it with 4 ml of digestion media containing 2% FBS, 100 ug/ml DNase I (Roche), and 100 ug/ml Liberase TM (Sigma-Aldrich) in DMEM. Tubes were moved to a 37°C water bath and fragments were triturated through a glass Pasteur pipette at 0 min and 6 min to mechanically aid digestion. At 12 min tubes were spun briefly to pellet undigested fragments and the supernatant was moved to 20 ml of 0.5% BSA (Sigma-Aldrich), 2 mM EDTA (TekNova), in PBS (MACS buffer) on ice to stop the enzymatic digestion. This was repeated twice for a total of three 12 min digestion cycles, or until there were no remaining tissue fragments. The single-cell suspension was then pelleted and washed once in MACS Buffer. Density-gradient centrifugation using a three-layer Percoll gradient (GE Healthcare) with specific gravities of 1.115, 1.065, and 1.0 was used to enrich

for stromal cells. Cells isolated from the Percoll-light fraction, between the 1.065 and 1.0 layers, were then resuspended in 0.5% BSA (Sigma-Aldrich), 2 mM EDTA (TekNova) (FACS buffer) and counted.

## Flow cytometry and cell sorting

Single-cell suspensions were prepared as described and incubated with Live/Dead Fixable Blue Dead Cell Stain (ThermoFisher) in PBS for 10 min at room temperature followed by blocking with anti-mouse CD16/CD32 (2.4G2) (UCSF Hybridoma Core Facility) and 5% normal rat serum for 10 min at room temperature. Cells were then washed in FACS buffer and stained for surface markers for 20 min at room temperature. For intracellular staining, cells were fixed and permeabilized using the eBioscience FoxP3 Transcription Factor Buffer Set (ThermoFisher) according to the manufacturer's instructions. Cells were either sorted directly into DMEM (ThermoFisher) containing 10% FBS using a BD FACS Aria Fusion (BD Biosciences) or analyzed using a BD LSR II (BD Biosciences) housed within the UCSF Single Cell Analysis Center. Flow cytometry data was analyzed using BD FACSDiva v8.0 or FlowJo v10.5.3 software (TreeStar Software). The following antibodies were used in this study: Ly51-PE (6C3, BioLegend), CD11c-PE-Cy7 (N418, eBioscience), CD45-PerCP (30-F11, BioLegend), EPCAM-APC-Cy7 (G8.8, BioLegend), Aire-e660 (5H12, eBioscience), Ki67-PE (eBioscience), CCL21 (59106, R and D Systems), Goat anti-Rat IgG-A488 (ThermoFisher).

## scRNA-seq

To prepare the cells for droplet-based sequencing, mTECs were flow-sorted into tubes containing 750 ul DMEM + 10%FBS (50,000 cells were sorted for the *Aire* trace experiment, 60,000 cells were sorted for the week two control, 37,000 cells were sorted for the week two experiment, 44,000 cells were sorted for the week four experiment, 32,000 cells were sorted for the week six experiment, 50,000 cells were sorted for the week 10 experiment, and 24,000 cells were sorted for the week 10 control). Cells were spun down at 300 g for 5 min and all but 50–100 ul of supernatant was removed to aim for a final concentration of 700–1000 cells/μl. An estimated 4000 single cells per sample were then subjected to 10x Genomics single-cell isolation and RNA-sequencing following the manufacturer's recommendations. Illumina HiSeq 4000 (Illuminia) was used for deep sequencing.

## Initial analysis of scRNA-seq data

Sequences from scRNA-seq were processed using Cellranger v2.2.0 software (*Zheng et al., 2017*). For the anti-RANKL experiment, sequences were processed using the cellranger mm10-1.2.0 genome and gtf file. For the *Aire* lineage tracing experiment, the sequence for the ZsGreen transcript was added to the fasta and the gtf file. A complete reference was made running cellranger mkref with the updated fasta and gtf files as arguments. For each sample, cell ranger mkfastq and cellranger count were run with the transcriptome argument pointing to the mm10 reference for the anti-RANKL experiment and the mm10 + ZsGreen reference for the lineage trace experiment.

Raw data generated by Cellranger were then read into the Seurat (*Butler et al., 2018*) v2.3.4 R package with at least 200 genes per cell and at least 3 cells. Cells were further filtered based on the number of genes per cell (between 200–7500) and the percent of mitochondrial reads per cell (0%–10%). The remaining cells and genes were used for downstream analysis. The data were normalized by using 'LogNormalize' method and data scaled with 'scale.factor=1000' from Seurat. For each sample, variable genes were found by using 'FindVariableGenes' with the following options mean. function = ExpMean, dispersion.function = LogVMR, x.low.cutoff = 0.0125, x.high.cutoff = 3, y. cutoff = 0.5.

## Analysis of controls

The week 2 and week 10 control samples were initially analyzed as explained in the section 'Initial analysis of scRNA-seq data'. After initial processing, the two samples were merged using the Merge-Seurat function. Variable genes were again found for the combined object as described above. The returned variable genes were used as the gene list given to the RunPCA function. Clusters were determined using the 'FindClusters' function with the following options reduction.type = 'pca', dims_use = 1:13, resolution = 0.6, random.seed = 0. Further dimensionality reduction was performed by using RunUMAP using the options reduction.type = 'pca', dims_use = 1:13. Markers of each cluster

were found using the 'FindMarkers' command and highly similar clusters were merged. A small cluster of cells resembling t-cells based on gene expression were removed from further analysis. Clusters were classified based on similarity of marker genes to the populations in the *Aire* lineage tracing experiment. Differential gene expression was performed by running FindMarkers on all pair-wise clusters (so that differentially expressed genes could be shared between clusters). Genes were called differentially expressed if the adjusted p-value was less than 0.05 and the log fold change was greater than 1. Differentially expressed genes were filtered to include only transcription factors using a list of mouse transcription factors (http://genome.gsc.riken.jp/TFdb/tf_list.html) and chromatin-modifying factors (*H2afz*, *Hmgb1*, *Hmgn1*, *H2afx*). These genes were manually searched for previous connections to mTEC development and *Aire* expression. Differentially expressed genes were then filtered to only include genes differentially upregulated in only the 'TAC-TEC' population, and not any other population. These genes were then compared to the list of genes differentially upregulated in cluster 5 and cluster 8 (identified as transit-amplifying populations) from *Basak et al., 2018*. To determine enrichment of these genes, we performed a hypergeometric test (overlapping genes = 21, genes in TAC-TEC = 35, genes in transit-amplifing clusters = 173, total genes = 20,309). The same p-value was determined using a fisher's exact test.

Cell-cycle state was determined using cyclone from scran v1.10.1 which uses a list of known cell-cycle genes to determine each cell a score for G2/M and G1 phase. Pairs of genes that change between cell-cycle phases (from *Scialdone et al., 2015*) are provided to cyclone. Scores are based on which gene within each pair is more highly expressed in each cell. These scores are used to assign each cell to G2/M, G1, or S phase.

## Trajectory analysis

Single-cell velocity trajectory analysis was performed in python using scvelo v0.1.24 from the Theis lab. First, a loom file were generated using run10x from velocyto (*La Manno et al., 2018*). The loom file was than loaded into python. Normalization and filtering was done using filter_and_normalize with the options min_counts = 20, min_counts_u = 10, n_top_genes = 3000. Next the moments were calculated using 30 PCs and 30 neighbors. Finally, velocity and the velocity graph were calculated. UMAP coordinates and cluster identification from the Seurat object were then added to the RNA velocity object. The velocity was plotted on the UMAP coordinates and colored by the cluster id. The scvelo output was used to determine lineage relationships and common transit-amplifying populations. Plots of RNA velocity were made using the command pl. velocity_embedding_stream from scvelo.

Pseudotime scores were determined using slingshot (*Street et al., 2018*) v1.1.0 with the 'TAC-TEC' cluster as the argument for start.clus. These pseudotime scores were used as the pseudotime values in *Figure 2E*. TSA genes were determined using the Fantom database and filtering out genes that were seen in five tissues or less (*Forrest et al., 2014*). Genes were further filtered to only keep protein-coding genes as described previously (*Brennecke et al., 2015*). *Aire*-dependent genes were defined based on the 2014 Sansom et al. list of *Aire*-dependent genes (*Sansom et al., 2014*). *Fezf2* dependent genes were identified by determining genes with a 2-fold enrichment in the control mouse vs. the *Fezf2* knockout mouse (*Takaba et al., 2015*). The percent of TSAs per cell was calculated by determining the number of TSAs expressed in a given cell and dividing by the total number of TSAs. TSA expression was averaged within each cell by dividing the total expression of TSAs by the total number of TSAs or the total number of expressed TSAs. Slingshot analysis was repeated after regenerating the dimensionally reduced object without any TSA genes included.

## Analysis of the Aire lineage tracing sample

For the *Aire* lineage tracing sample, the returned variable genes were used as the gene list given to the RunPCA function. Clusters were determined using the 'FindClusters' function with the following options reduction.type = 'pca', dims_use = 1:12, resolution = 0.6, random.seed = 0. FindClusters embeds cells in a k-nearest neighbor graph with edges drawn between cells with similar gene expression. These are these partitioned into highly inter connected communities ased on Euclidean distance in PCA space. Further dimensionality reduction was performed by using RunUMAP using the options reduction.type = 'pca', dims_use = 1:12. Markers of each cluster were found using the 'FindMarkers' command and highly similar clusters were merged. A small cluster of cells resembling

t-cells based on gene expression were removed from further analysis. Clusters were classified based on common marker genes previously described for each mTEC population. Clusters highly expressing *Aire* were further characterized by their expression of ZsGreen. This analysis was repeated after cell-cycle regression to ensure conclusions were robust (Data not shown). Double positive percents were found by using a cutoff of 0 for ZsGreen, *Aire, Ackr4, Trpm5, and Fezf2*, a cutoff of 2 for *Krt10* and a cutoff of 4 for *Ccl21a*.

Cell-cycle state was again determined using cyclone from scran v1.10.1. All results in the controls replicated observations made from the control mice (*Figure 3—figure supplement 1*).

## Analysis of all samples

All samples were processed as described in the 'Initial analysis of scRNA-seq data'. The top 1000 most highly variable genes from each sample were merged, keeping only genes that were present in all samples were used for CCA analysis. The 7 Seurat objects and the variable genes found above were used to generate a new Seurat object with the RunMiltiCCA function, using num.ccs = 30. The canonical correlation strength was calculated using num.dims = 1:30 and the samples were aligned using dims.align = 1:20. Clusters were found using the 'FindClusters' function based on the aligned CCA with the following options reduction.type = ('cca.aligned', resolution = 0.6, dims.use = 1:20, random.seed = 0). Further dimensionality reduction was performed by using RunUMAP using the options reduction.type = 'cca.aligned', dims_use = 1:20, seed.use = 0. The cluster identities were assigned based on the *Aire* trace experiment. The *Aire* trace cluster identities were added to the Seurat object containing all samples. For each cluster, the identity was assigned based on which *Aire* trace population it shared the most cells with (*Figure 4b*). There was a clear majority of cells aligning with the *Aire* trace populations for all clusters. Population assignments were checked based on gene expression of marker genes such as *Ackr4, Ccl21a, Aire, Krt10*, and *Trpm5* by plotting expression of marker genes on the UMAP dimensional reduction (*Figure 4—figure supplement 1*), and by plotting heatmaps of marker genes separated by population for each sample (*Figure 4—figure supplement 3*). To ensure that the cells mapping to the '*Aire*-positive' cluster were correctly mapping to the cluster we looked at expression of the tope 30 genetic markers of the '*Aire*-positive' cluster (using FindMarkers in the wild-type samples) (*Figure 6—figure supplement 1*). A small cluster of cells resembling t-cells based on gene expression were removed from further analysis. Cell-cycle state was again determined using cyclone from scran.

Pseudotime scores were determined using slingshot (*Street et al., 2018*) v1.1.0 with the 'TAC-TEC' cluster as the argument for start.clus. These pseudotime scores were used as the pseudotime values in *Figure 4E*. The density of cells across psuedotime for each sample was plotted using ggplot2.

## Further analysis of 'TAC-TEC' population

Cells in the 'TAC-TEC' population for each sample were subset from the Seurat of all samples using SubsetData with subset.raw = TRUE. Data from each seven new Seurat objects were normalized by using 'LogNormalize' method and data scaled with 'scale.factor=1000' from Seurat. For each sample, variable genes were found by using 'FindVariableGenes' with the following options mean. function = ExpMean, dispersion.function = LogVMR, x.low.cutoff = 0.0125, x.high.cutoff = 3, y.cutoff = 0.5. The top 1000 most highly variable genes from each sample were merged, keeping only genes that were present in all samples were used for CCA analysis. The 7 Seurat objects and the variable genes found above were used to generate a new Seurat object with the RunMiltiCCA function, using num.ccs = 10. The canonical correlation strength were calculated using num.dims = 1:10 and the samples were aligned using dims.align = 1:10. Clusters were found using the 'FindClusters' function based on the aligned CCA with the following options reduction.type = ('cca.aligned', resolution = 0.7, dims.use = 1:7, random.seed = 0). Further dimensionality reduction was performed by using RunUMAP using the options reduction.type = 'cca.aligned', dims_use = 1:10, seed.use = 0. Differential gene expression was performed by running FindAllMarkers. Genes were called differentially expressed if the adjusted p-value was less than 0.05 and the log fold change was greater than 0.5.

## UMI correction and cell down-sampling

A Seurat object containing only cells from the 'Aire-positive' population was created using Subset-Data. UMIs for each sample were down-sampled to the number of UMIs in the week four sample using DropletUtils v1.2.1 downsampleMatrix (Griffiths et al., 2018; Lun et al., 2019). Effectiveness of the down-sampling was determined by observing the number of dropouts of housekeeping genes before and after the down-sampling (Figure 6—figure supplement 3d). The percent of dropouts was computed by dividing the number of cells with expression values of 0 for each housing keeping gene (Chmp2a, Emc7, Psmb4, Vcp, and Gapdh) by the total number of cells for each sample. This was calculated before and after down-sampling UMIs for each sample.

Cumulative fraction of genes was determined for TSAs, Aire-dependent genes, Fezf2-dependent genes, and all other genes (genes not listed as TSAs, Aire-dependent, or Fezf2-dependent). For each gene list, a gene was counted as 'expressed' if at least one cell in a sample had an expression value greater than 0. The percent of the gene list expressed was determined by dividing the total number of expressed genes by the total number of genes in the list. The cumulative fraction of genes detected was also determined for the control samples before downsampling UMIs to determine the number of genes detected in these samples (Figure 6—figure supplement 3A). To determine the number of genes expressed per cell from each list, a gene was again counted as 'expressed' if its expression was greater than 0 for the given cell.

To determine the number of genes from each gene set expected for a specific number of cells from the 10 week control 'Aire-positive' population, cell_sampler from dropseqr (https://rdrr.io/github/argschwind/dropseqr/) was used with ncells set to the number of cells in the 'Aire-positive' population in each sample. For each subsample of the control, the percent of observed genes from each gene set was calculated 1000 times. The percent values were used to create a distribution for each sample. The actual observed percent of the gene set was plotted as a red line and a p-value was determined using a z-score using the mean and standard deviation from the distribution of sub-sampled cells from the control (Figure 6—figure supplement 3D).

## Gene sets

The list of Aire-dependent genes was originally determined by comparing Aire knockout to WT mTECs (Sansom et al., 2014). Fezf2-dependent genes were determined by performing differential expression of a Fezf2 knockout to WT mTECs from a published dataset using a adjusted p-value cutoff of 0.05 and a log fold change value of 1 (Takaba et al., 2015). TSA genes were found using the Fantom database and curated as previously described (Brennecke et al., 2015; Forrest et al., 2014). Genes were called a TSA if they were observed in five tissues or less. All other genes were determined by taking all protein-coding genes and subtracting out genes from the Aire-dependent, Fezf2-dependent, and TSA gene lists.

## Acknowledgements

We thank Sandy Klemm, Elisa Zhang, Mariona Nadal Ribelles, and Michael Sikora for helpful feedback and discussion. This work was supported by an NSF Graduate Research Fellowship to KLW under Grant No. DGE- 1656518 and NIH grant P01 HG000205 to LMS. This study used the Genome Sequencing Service Center from the Stanford Center for Genomics and Personalized Medicine Sequencing Center, supported by NIH S10OD020141, and computing resources provided by the Stanford Genetics Bioinformatics Service Center. This work was also supported by NIH grants R01 AI097457 and R01 AI097457 to MSA NIH and used the UCSF Single Cell Analysis Center, supported by NIH 1S10OD021822-01. Figures 1a and 3a were generated with BioRender.com.

## Additional information

### Funding

| Funder | Grant reference number | Author |
| --- | --- | --- |
| National Science Foundation | DGE- 1656518 | Kristen L Wells |
| National Institutes of Health | P01 HG000205 | Lars M Steinmetz |

| National Institutes of Health | R01 AI097457 | Mark S Anderson |

The funders had no role in study design, data collection and interpretation, or the decision to submit the work for publication.

## Author contributions

Kristen L Wells, Conceptualization, Resources, Data curation, Software, Formal analysis, Validation, Investigation, Visualization, Methodology, Writing - original draft, Project administration, Writing - review and editing; Corey N Miller, Conceptualization, Resources, Data curation, Formal analysis, Validation, Investigation, Visualization, Methodology, Writing - original draft, Writing - review and editing; Andreas R Gschwind, Resources, Supervision; Wu Wei, Supervision; Jonah D Phipps, Investigation; Mark S Anderson, Conceptualization, Resources, Supervision, Funding acquisition, Writing - original draft, Project administration, Writing - review and editing; Lars M Steinmetz, Conceptualization, Supervision, Funding acquisition, Writing - original draft, Project administration, Writing - review and editing

## Author ORCIDs

Kristen L Wells ◎ https://orcid.org/0000-0002-7466-8164
Corey N Miller ◎ https://orcid.org/0000-0002-1291-0074
Andreas R Gschwind ◎ http://orcid.org/0000-0002-0769-6907
Mark S Anderson ◎ https://orcid.org/0000-0002-3093-4758
Lars M Steinmetz ◎ https://orcid.org/0000-0002-3962-2865

## Ethics

Animal experimentation: Mice were maintained in the University of California San Francisco (UCSF) specific pathogen- free animal facility in accordance with the guidelines established by the Institutional Animal Care and Use Committee (IACUC) and Laboratory Animal Resource Center and all experimental procedures were approved by the Laboratory Animal Resource Center at UCSF. The animal protocol number associated with the study is AN180637-02B.

## Decision letter and Author response

Decision letter https://doi.org/10.7554/eLife.60188.sa1
Author response https://doi.org/10.7554/eLife.60188.sa2

# Additional files

## Supplementary files

• Supplementary file 1. Frequencies of cells from each population based on expression of zsGreen and marker genes of each population (*Aire*, *Ackr4*, *Trpm5*, *Fezf2*, *Krt10*, and *Ccl21a*). Positive expression of each gene was based on clear separations between populations (zsGreen: 0, *Aire*: 0, *Krt10*: 2, *Ackr4*: 0, *Trpm5*: 0, *Ccl21a*: 4).

• Transparent reporting form

## Data availability

RNA-seq data that support the findings of this study have been deposited in the GEO database under accession numbers GSE137699.

The following dataset was generated:

| Author(s) | Year | Dataset title | Dataset URL | Database and Identifier |
|---|---|---|---|---|
| Wells KL, Miller CN, Gschwind AR, Phipps JD, Anderson MS, Steinmetz LM | 2020 | Combined transient ablation and single cell RNA sequencing reveals the development of medullary thymic epithelial cells | https://www.ncbi.nlm.nih.gov/geo/query/acc.cgi?acc=GSE137699 | NCBI Gene Expression Omnibus, GSE137699 |

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
