## [Decision Letter]

**Acceptance summary:**

In this manuscript the authors combine single cell RNAseq analyzes with genetic cell fate mapping and transient ablation to examine the developmental relationship between thymic epithelial cells in adult mice. The manuscript offers new insights into the role of RANK signaling in mTEC differentiation, and provides validation of previously identified thymic epithelial cell populations and their developmental relationships. They also provide intriguing information about the kinetics of expression of the *AIRE* and *Fezf2* transcriptional regulators and the tissue restricted antigen (TSA) that they regulate.

**Decision letter after peer review:**

Thank you for submitting your article "Combined transient ablation and single cell RNA-sequencing reveals the development of medullary thymic epithelial cells" for consideration by *eLife*. Your article has been reviewed by three peer reviewers, one of whom is a member of our Board of Reviewing Editors, and the evaluation has been overseen by Tadatsugu Taniguchi as the Senior Editor. The reviewers have opted to remain anonymous.

The reviewers have discussed the reviews with one another and the Reviewing Editor has drafted this decision to help you prepare a revised submission.

Summary:

In this manuscript the authors combine single cell RNAseq analyzes with genetic cell fate mapping and transient ablation to examines the developmental relationship between thymic epithelial cells in adult mice. The manuscript is well written and provides further validation of previously published data and offers new insights into the role of RANK signaling in mTEC differentiation. Another intriguing observation is the delay between AIRE induction and Aire-dependent tissue restricted antigen (TSA) expression, and the distinct kinetics displayed by *Fezf2* and its targets. Unfortunately, the authors do not appear to acknowledge all essential prior work that described similar populations and arrived at similar conclusions. There are also some key technical issues that should be corrected.

Essential revisions:

1) The TAC-TEC population appears to be the same as or very similar to the proliferating mTEC cluster characterized in Dhalla et al., 2020. While this earlier paper is cited in the Introduction, the authors should acknowledge that this past work identified a proliferating mTEC population that was predicted to give rise to both Ccl21-expressing mTEC (termed pre-Aire by those workers, possibly incorrectly) and Aire+ mTEC, by very similar bioinformatic approaches (Velocity).

2) One of the authors' major conclusions, that TAC-TECs give rise to both the Ccl21a-expressing branch and the AIRE expression branch of mTEC development, is not strongly supported by the current data. In particular, there is a possibility the TAC-TEC cluster contains with separate and developmentally unrelated precursors for Aire+ and Ccl21a-hi (and Tufts cell) mTEC populations. Given that substantial fraction of TAC-TECs is characterized by expression of Aire mRNA (reflecting an active Aire locus) (Figure 1C, Figure 1—figure supplement 1C) and that a substantial fraction of this subset even expresses the Aire protein (Figure 2C) – one would expect a large fraction of the Ccl21 cluster to also be ZsGreen positive (like late mTECs or tuft cells). However Figure 3C, shows almost no expression of ZsGreen reporter in the Ccl21+ cluster. The authors also argue for their conclusion based on the co-expression of Ccl21 protein and ZsGreen reporter. "Approximately half of cells that expressed CCL21 protein also expressed ZsGreen reporter but 25% of these double-positive cells were MHCIIlo, indicating that a subset of “Ccl21a-high” cells are downstream of an Aire-expressing population (Figure 3E and Figure 3—figure supplement 1E). The finding that Ccl21a cells are marked by ZsGreen is consistent with the pseudotime prediction that the “Ccl21a-high” population is downstream of the “TAC-TEC” population rather than the progenitor of the “Aire-positive” population". However, these data appear to be line with previous findings suggesting that Ccl21+ cells (jTECs/mTEC-I) give rise to Aire+ cells. The likely explanation for this is that the Ccl21 protein persists into Aire+ stage (Lkhagvasuren and Kozai et al.) rather than the other way around. Moreover, there is evidence from other groups showing that mTEClo population in the adult thymus contains progenitors that give rise to mTEChi cells. If the mTEClo population consists of tuft cells, late mTECs and Ccl21+ mTECs, which of these cells gives rise to mTEChi cells? Ideally, the authors should define the TAC-TECs by unique surface markers, which would allow them to isolate this population and experimentally validate some of their conclusions. In the absence of clearer experimental evidence regarding the developmental potential of the 2 TAC-TEC populations, the authors should discuss alternative interpretations of their findings.

3) More characterization of the clusters is needed. For example, what proportion of the Ccl21a-hi cluster cells were ZsGreen+ (the text only specifies a "small number")? Given that chemokines are secreted and present on surfaces of other cells, these double positive cells should be isolated, and Ccl21 expression verified by qPCR. What is the MHC II profile on the Ccl21+, Aire/lineage negative populations in Figure 3E and Figure 4D? While in Figure 1 the authors show that the TAC-TEC express low levels of Aire and *Fezf2* mRNA – the authors do not show other key developmental markers that traditionally define the mTEC compartment and its developmental stages. The authors should also include MHCII, CD40, CD80 or RANK (which becomes very relevant in the second part of the manuscript). Similarly, the authors should also include additional markers that were previously shown to define the mTECI and/or jTEC populations, such as Itgb4, Itga6, Sox4, Pdpn, etc.

4) Were genes considered as TSAs excluded from the list of ordering genes used to calculate the pseudotime trajectory? If this was not done, do the authors know if conclusions remain evident when the ordering genes are distinct from known TSAs?

---

## [Author Response]

Essential revisions:1) The TAC-TEC population appears to be the same as or very similar to the proliferating mTEC cluster characterized in Dhalla et al., 2020. While this earlier paper is cited in the Introduction, the authors should acknowledge that this past work identified a proliferating mTEC population that was predicted to give rise to both Ccl21-expressing mTEC (termed pre-Aire by those workers, possibly incorrectly) and Aire+ mTEC, by very similar bioinformatic approaches (Velocity).

We agree this recent work should be more forcefully acknowledged and have added this to the Discussion:

“Other recent single cell studies (Dhalla et al., 2020) have also identified a proliferating mTEC population that is very similar to the “TAC-TEC” population presented here. […] Using diffusion pseudotime, the authors identified similar trajectories to our analysis but concluded that proliferating mTECs were downstream of the Ccl21a-expressing mTECs based on their RNA velocity data (Dhalla et al., 2020).”

2) One of the authors' major conclusions, that TAC-TECs give rise to both the Ccl21a-expressing branch and the AIRE expression branch of mTEC development, is not strongly supported by the current data. In particular, there is a possibility the TAC-TEC cluster contains with separate and developmentally unrelated precursors for Aire+ and Ccl21a-hi (and Tufts cell) mTEC populations. Given that substantial fraction of TAC-TECs is characterized by expression of Aire mRNA (reflecting an active Aire locus) (Figure 1C, Figure 1—figure supplement 1C) and that a substantial fraction of this subset even expresses the Aire protein (Figure 2C) – one would expect a large fraction of the Ccl21 cluster to also be ZsGreen positive (like late mTECs or tuft cells). However Figure 3C, shows almost no expression of ZsGreen reporter in the Ccl21+ cluster.

We agree that additional visualizations should be added in addition to the violin plot in Figure 3C to better represent zsGreen transcript levels. zsGreen reporter protein is readily detected in about half of Ccl21 protein-expressing cells by flow cytometry. Therefore, in zsGreen+ cells recombination of the reporter locus has definitively occurred and, given the mechanism of the lineage-tracing system, transcription will be driven by the constitutive CAG promoter and transcript should be present. To address this, we have provided an alternative visualization of zsGreen transcript that confirms it is detected in approximately 34% of the “Ccl21a-high” cluster. We have also included a table that indicates what percent of each population expresses all of the genes shown in the new visualization. Additionally, this analysis shows that 18% of the TAC-TEC cluster is Aire-, ZsGreen- indicating that some of the cells feeding the Ccl21a population may never pass through an Aire-expressing stage. We have added this data to the manuscript as Figure 3—figure supplement 2 and Supplementary file 1 to provide improved clarity for the reader. We have updated the subsection “Aire lineage tracing mice demonstrate cellular relationships in the Aire branch of mTEC development” to include percentages relating to the expression in each population.

Finally, the distribution of lineage reporter may be related to the kinetics of cell turnover in the medullary compartment. For example, it has previously been shown that after 3 weeks of BrdU labeling, 80% of mTEC-hi cells were post-mitotic and only 30% of mTEC low cells were post-mitotic. Therefore, the Ccl21+ cells may have lower ZsGreen expression than, for example the late tuft cells, because a smaller proportion of the “Ccl21a-high” population has turned over and differentiated during the 10-day tamoxifen treatment.

The authors also argue for their conclusion based on the co-expression of Ccl21 protein and ZsGreen reporter. "Approximately half of cells that expressed CCL21 protein also expressed ZsGreen reporter but 25% of these double-positive cells were MHCIIlo, indicating that a subset of “Ccl21a-high” cells are downstream of an Aire-expressing population (Figure 3E and Figure 3—figure supplement 1E). The finding that Ccl21a cells are marked by ZsGreen is consistent with the pseudotime prediction that the “Ccl21a-high” population is downstream of the “TAC-TEC” population rather than the progenitor of the “Aire-positive” population". However, these data appear to be line with previous findings suggesting that Ccl21+ cells (jTECs/mTEC-I) give rise to Aire+ cells. The likely explanation for this is that the Ccl21 protein persists into Aire+ stage (Lkhagvasuren and Kozai et al.) rather than the other way around.

We agree that the flow cytometry data alone is not definitive evidence of the precursor-product relationship proposed in the manuscript. Instead, this model is constructed by considering the totality of both bioinformatic and wet lab findings. Indeed, examination of ZsGreen positive cells in the “Ccl21a-high” population has shown that only 2% of the ZsGreen positive cells also express Aire. Given the kinetics of this reporter system (in which ZsGreen lags behind Aire), this observation provides strong additional evidence arguing against Ccl21 expression persisting into the Aire+ stage and is instead suggestive of a population in which Aire was previously expressed (post-Aire).

Moreover, there is evidence from other groups showing that mTEClo population in the adult thymus contains progenitors that give rise to mTEChi cells. If the mTEClo population consists of tuft cells, late mTECs and Ccl21+ mTECs, which of these cells gives rise to mTEChi cells? Ideally, the authors should define the TAC-TECs by unique surface markers, which would allow them to isolate this population and experimentally validate some of their conclusions. In the absence of clearer experimental evidence regarding the developmental potential of the 2 TAC-TEC populations, the authors should discuss alternative interpretations of their findings.

While we identify a population of cells highly enriched for actively cycling transit-amplifying TECs, we do not purport to identify stem-like progenitors. In Figure 2C of the manuscript, we show that surface levels of MHCII on Ki67bright TECs are considerably lower (on a log scale) than those on bona fide MHCIIhi cells. Furthermore, there is variability in the MHCII surface levels on Ki67bright TECs and some of these cells are definitively MHCIIlo. Finally, the levels of transcript for MHCII alleles within the TAC-TEC population are low compared to “Aire-positive” mTECs (see below, point 3 and new Figure 1C). Therefore, taken together, we believe these observations are fully consistent with previous reports that mTEClo (a broad concept dependent on the technical details of experimental design and data analysis), are likely to contain at least some Ki67bright cycling TECs and may also include even earlier stem-like progenitors not described here.

We agree that unique surface markers of the TAC-TEC population are critical for more robust characterization of this population. Despite significant efforts to identify such markers, putative markers we have evaluated to date have proved unsuitable (data not shown) and this work remains ongoing. Therefore, we believe that surface characterization of TAC-TECs is beyond the scope of the current manuscript and hope to provide such markers in future publications.

Finally, to underscore and acknowledge alternative interpretations of our data, we have added new text to the Discussion:

“Furthermore, Gray et al. demonstrated that mTEC-lo can give rise to Aire-expressing mTEC-hi with low efficiency, suggesting only a minor subset of mTEC-lo maintain this potential (Gray et al., 2007). […] We therefore speculate that the mTEC-lo cells described by Gray et al., which eventually give rise to Aire-expressing mTECs, are in fact proliferating “TAC-TECs”.”

“While we combine multiple experimental techniques to support the branching model proposed here, aspects of both models could fit our data. […] Further validation including lineage tracing using methods such as SCAR-trace (Raj et al., 2018; Spanjaard et al., 2018) or isolation of the TAC-TEC population for evaluation of their ex-vivo differentiation capacity will be required to confirm which model is correct.”

3) More characterization of the clusters is needed. For example, what proportion of the Ccl21a-hi cluster cells were ZsGreen+ (the text only specifies a "small number")?

We believe the improved visualization of zsGreen transcript in approximately 34% of the “Ccl21a-high” cluster, as well as in a subset of “Aire-positive” and “Late-Aire” cells directly address this comment (please see Discussion and data above under Essential revisions 2).

Given that chemokines are secreted and present on surfaces of other cells, these double positive cells should be isolated, and Ccl21 expression verified by qPCR.

We agree that surface binding of chemokines could account for Ccl21 signal by flow cytometry and confound interpretation—especially of “double positive” events. To evaluate this possibility, we performed Ccl21 intracellular staining with and without intracellular permeabilization. The data in Figure 3—figure supplement 3 is gated on CD11c- EpCam+ CD45- TECs. Notably, even in the no permeabilization control, a small amount of Aire staining is still present. Therefore, setting the Ccl21 gate using the no permeabilization control is expected to provide an underestimate of true intracellular staining. However, using this stringent gating strategy, we still see events that are both Aire+ and Ccl21+, demonstrating that at least a subset of “double positive” cells contain intracellular protein and supporting our scRNA-seq observation of rare Aire and Ccl21 co-expressing cells. This data has been added to Figure 3—figure supplement 3C.

What is the MHC II profile on the Ccl21+, Aire/lineage negative populations in Figure 3E and Figure 4D? While in Figure 1 the authors show that the TAC-TEC express low levels of Aire and Fezf2 mRNA – the authors do not show other key developmental markers that traditionally define the mTEC compartment and its developmental stages. The authors should also include MHCII, CD40, CD80 or RANK (which becomes very relevant in the second part of the manuscript). Similarly, the authors should also include additional markers that were previously shown to define the mTECI and/or jTEC populations, such as Itgb4, Itga6, Sox4, Pdpn, etc.

We have provided MHCII profiles for the Ccl21+ subset to Figure 3—figure supplement 1F and Figure 4—figure supplement 2B.

We agree that more mTEC markers would be valuable to the interpretation of the clustering analysis. Therefore, we have generated a heatmap using a curated list of key genes from the literature described in both low dimensional mTEC analysis and mTEC single cell analysis. We have added this data to the manuscript as Figure 1C. This heatmap includes genes used in previous studies to identify populations. The inclusion of these genes will allow the populations described in our study to be more easily compared to existing mTEC publications. Additionally, we have added MHCII genes (H2-Aa and H2-Eb1). At the RNA level, the expression of MHCII genes agrees with our analysis using flow cytometry, that the expression of MHCII by the “TAC-TEC” population is higher than the “Ccl21a-high” population and lower than the “Aire-positive” and “Late-Aire” populations. Finally, we added Pdpn, Itga6, Sox4, and Itgb4 as requested. All of these genes were primarily expressed in the “Ccl21a-high” and “cTEC” populations.

Finally, we examined the expression of Pdpn (gp38 by flow cytometry). Similar to the findings of Onder et al., we do see a small population of Pdpn+ EpCAM+ TECs that are MHCIIlo (see Author response image 1). However, we find that the frequency of Pdpn+ TECs is quite low and that the vast majority of Ki67bright TECs are Pdpn-. Given the lack of definitive surface markers for the “TAC-TEC” population and the uncertainty of the relationship between Pdpn and “TAC-TECs”, we feel that a detailed discussion of Pdpn within the TEC compartment is beyond the scope of the current manuscript.

4) Were genes considered as TSAs excluded from the list of ordering genes used to calculate the pseudotime trajectory? If this was not done, do the authors know if conclusions remain evident when the ordering genes are distinct from known TSAs?

Thank you for this excellent suggestion! There were a small number of TSAs included in the ordering genes used to calculate the pseudotime trajectory. We have repeated the analysis without the TSAs included and it does not impact our results (and have added this data as Figure 2—figure supplement 2C). We have added a sentence reflecting this addition to the subsection “TSAs appear late in mTEC development”.